# A top-down approach of surface carbonyl sulfide exchange by a Mediterranean oak forest ecosystem in Southern France

S. Belviso[1], I. M. Reiter[2,3], B. Loubet[4], V. Gros[1], J. Lathière[1], D. Montagne[4], M. Delmotte[1], M. Ramonet[1], C. Kalogridis[1], B. Lebegue[1], N. Bonnaire[1], V. Kazan[1], T. Gauquelin[5], C. Fernandez[5], and B. Genty[3]

[1]Laboratoire des Sciences du Climat et de l'Environnement, LSCE/IPSL, CEA-CNRS-UVSQ, Université Paris-Saclay, F-91191 Gif-sur-Yvette, France
[2]CNRS, FR 3098 ECCOREV, Europôle de l'Arbois, F-13545 Aix-en-Provence, France
[3]CEA, CNRS, Aix-Marseille University, UMR 7265 Biologie Végétale et Microbiologie Environnementales, F-13115 Saint Paul-lez-Durance, France
[4]AgroParisTech, INRA, Université Paris-Saclay, UMR 1402 Ecosys, 78 850 Thiverval-Grignon, France
[5]Aix Marseille Univ, Avignon Université, CNRS, IRD, IMBE Institut Méditerranéen de Biodiversité et d'Ecologie marine et continentale, Marseille, France
*Correspondence to*: Sauveur Belviso (sauveur.belviso@lsce.ipsl.fr)

**Abstract.** The role that soil, foliage and atmospheric dynamics have on surface carbonyl sulfide (OCS) exchange in a Mediterranean forest ecosystem in Southern France (the Oak Observatory at the Observatoire de Haute Provence, O3HP), was investigated in June of 2012 and 2013 with essentially a top-down approach. Atmospheric data suggest that the site is appropriate for estimating gross primary production (GPP) directly from eddy covariance measurements of OCS fluxes, but is less adequate for scaling net ecosystem exchange (NEE) to GPP from observations of vertical gradients of OCS relative to $CO_2$ during daytime. Firstly, OCS and carbon dioxide ($CO_2$) diurnal variations and vertical gradients show no net exchange of OCS during the night when the carbon fluxes are dominated by ecosystem respiration. This contrasts with other oak woodland ecosystems of a Mediterranean climate, where nocturnal uptake of OCS by soil and/or vegetation has been observed. Since temperature, the water and organic carbon content of soil at the O3HP should favor the uptake of OCS, the lack of nocturnal net uptake would indicate that its gross consumption in soil is compensated by emission processes that remain to be characterized. Secondly, the uptake of OCS during the photosynthetic period was characterized in two different ways. We measured ozone ($O_3$) deposition velocities and estimated the partitioning of $O_3$ deposition between stomatal and non-stomatal pathways before the start of a joint survey of OCS and $O_3$ surface concentrations. We observed an increasing trend in the relative importance of the stomatal pathway during the morning hours and synchronous steep drops of mixing ratios of OCS (amplitude in the range of 60-100 ppt) and $O_3$ (amplitude in the range of 15-30 ppb) after sunrise and before the break-up of the nocturnal boundary layer. The uptake of OCS by plants was also characterized from vertical profiles. However, the time window for calculation of the ecosystem relative uptake (ERU) of OCS, which is a useful tool to partition measured NEE, was limited in June 2012 to few hours after midday. This is due to the disruption of the vertical distribution of OCS by entrainment of OCS rich tropospheric air in the morning, and as the vertical gradient of $CO_2$ reverses when it is

still light. Moreover, polluted air masses (up to 700 ppt of OCS) produced dramatic variation in atmospheric OCS-to-$CO_2$ ratios during daytime in June 2013, further reducing the time window for ERU calculation.

## 1 Introduction

Terrestrial ecosystems modulate the water balance over land and fix carbon dioxide ($CO_2$) from the atmosphere in the form of carbon rich materials. Experimental and modeling studies have shown that changes in atmospheric $CO_2$ concentration and changes in climate, induced by increasing anthropogenic emissions of greenhouse gases, impact on the fixation of atmospheric $CO_2$ by plants (gross primary production, GPP), and on the release of $CO_2$ by terrestrial ecosystems (respiration, Reco) as modulated by temperature and water availability, and effects by fertilization (.g. Arora and Boer, 2014). Large

uncertainties in the determination in GPP and Reco fluxes at the continental scale and in the magnitude of effects induced by climate and fertilization remain. Further experimental and modeling studies should help to better constrain those fluxes.

In the late 80's, vegetation has been proposed to be the missing sink in the global cycle of atmospheric carbonyl sulfide (OCS; Brown and Bell, 1986; Goldan et al., 1988) and the first evidence from field observations of the uptake of OCS near the ground was provided by Mihalopoulos et al. (1989). Nowadays, the mechanistic link between leaf $CO_2$ and OCS

exchange is well understood (Stimler et al., 2010; Seibt et al., 2010; Wohlfahrt et al., 2012) and the scientific community has reached consensus on the potential of atmospheric OCS measurements to provide independent constraints on GPP at canopy (Blonquist et al., 2011; Asaf et al., 2013), regional (Campbell et al., 2008) and global (Montzka et al., 2007; Berry et al., 2013; Launois et al., 2015) scales. However, recent studies  also demonstrated limitations to the use of OCS as a GPP proxy at canopy and ecosystem scales because (1) consumption and/or production of OCS occur in soil and litter (Van Diest and

Kesselmeier 2008; Sun et al., 2015; Ogée et al., 2016; Whelan et al., 2016 and references therein), (2) in agricultural fields and midlatitude forests OCS can be taken up by plants also during the night (Maseyk et al., 2014; White et al., 2010; Commane et al., 2015), and (3) the leaf relative uptake of OCS and of $CO_2$ (LRU), which is of central importance in the calculation of GPP from eddy covariance measurements of OCS exchange ($L_{OCS}$) following Eq. (1), exhibit  daily and seasonal variations of variable amplitudes (Berkelhammer et al., 2014; Maseyk et al., 2014; Commane et al., 2015).

$$GPP = (L_{OCS} / LRU).([CO_2] / [OCS]) \tag{1}$$

The character L in $L_{OCS}$ stands for leaf because OCS exchange equals $L_{OCS}$ when other ecosystem fluxes are negligible. To address the diel LRU variations and the role of soil and litter for canopy scale analysis, some research groups are now combining canopy flux, leaf and soil chamber measurements in the field (L. Kooijmans personal communication, Sep. 2016).

Eq. 1 can also be used for regional scale analysis (Campbell et al., 2008). At this scale, LRU also varies as a function of plant type (i.e. C3 vs. C4 plants, Stimler et al., 2011). However, Hilton et al. (2015) demonstrated that the effect of LRU

variability was less significant at regional than at canopy scale because the regional spatial uncertainty in GPP is much larger than the LRU uncertainty.

The use of leaf and soil chambers offers a means to investigate in laboratory and field conditions the ability of plants and soils to degrade ambient OCS (e.g. Stimler et al., 2010; Sun et al., 2015). Approaches which avoid manipulation of biological material, such as the eddy flux, gradient or Radon-tracer methods (e.g. Maseyk et al., 2014; Commane et al., 2015; Belviso et al., 2013), can document over short and long time-spans the direction and the magnitude of surface OCS exchange at the ecosystem level. At continental or global scales, biosphere-atmosphere fluxes can be assessed from dynamic global vegetation models and all flux components can be optimized using satellite or global network data (e.g. Berry et al., 2013; Launois et al., 2015; Kuai et al., 2015). The global network NOAA ESRL for measurements of greenhouse gases in the atmosphere monitors OCS mixing ratios on a weekly basis since year 2000 (Montzka et al., 2007). It is in this framework that the major role of vegetation in the global budget of OCS was again emphasized. A second network (AGAGE) exists where air samples are analyzed every 60 minutes, but OCS data are not yet available for public access. Other sites have recently been instrumented for long term monitoring of atmospheric OCS concentrations and/or fluxes. They include a mixed temperate forest in North America (Harvard forest (Commane et al., 2015)), a boreal pine forest of South Finland (Hyytiälä, A. Praplan, personal communication, 2015) and a station located on the northern coast of the Netherlands (Lutjewad, H. Chen, personal communication, 2014; Kooijmans et al., 2016). Although rural and sub-urban areas have also been instrumented for shorter periods (Berkelhammer et al., 2014; Belviso et al., 2013 and references therein), yet many biomes remain unexplored. In summer 2012 and 2013, we used the facilities of the experimental field site Oak Observatory at the Observatory of the Haute Provence (O3HP), Saint Michel l'Observatoire, France, to study the biosphere-atmosphere exchanges of three atmospheric compounds (OCS, $CO_2$ and ozone ($O_3$)) which share stomatal uptake as a common pathway. O3HP is a Mediterranean forest ecosystem with low canopy height, and dominated by deciduous Downy oak *Quercus pubescens* Willd and Montpellier Maple, *Acer monspessulanum*. Often occurring in the transition of climate zones from Mediterranean to sub-Mediterranean, and thus potentially rather sensitive and responsive to climate change, *Q. pubescens* is an interesting model to monitor changes affecting the Mediterranean forest ecosystems.

Our top-down approach, similar to the approach by Blonquist et al. (2011), aims at determining the role of soil, foliage, atmospheric dynamics and air pollution on surface OCS exchange at the O3HP, at finding consistencies and differences with other oak woodland ecosystems characterized by a Mediterranean climate, and at assessing the desirability to use OCS to partition $O_3$ deposition between stomatal and non-stomatal pathways. Since direct LRU and OCS flux measurements were not performed during the campaigns, we used the ecosystem relative uptake (ERU) approach of Campbell et al. (2008) to provide a rough estimation of LRU variations using the following equation:

$$LRU = [ERU].[NEE / GPP] \tag{2}$$

where ERU is the relative gradient of OCS ($m^{-1}$) divided by the relative gradient of $CO_2$ ($m^{-1}$) and NEE is the net ecosystem exchange of $CO_2$ from eddy covariance measurements carried out at the site.

## 2 Material and methods

### 2.1 Description of the site and of air circulation

The two campaigns took place in June of 2012 and 2013. Both were of short duration (i.e. about two-weeks long). A description of the O3HP site is available in Kalogridis et al. (2014) and Santonja et al. (2015). In short, the site (43.93 N, 5.71 E) is located on the premises of Observatoire de Haute Provence, about 60 km north of Marseille, France, at an elevation of 680m above mean sea level. It is implemented in a forest area that has remained untouched at the least since 1945. The climate is sub-Mediterranean with warm-to-hot, and dry summers.

The O3HP observatory is characterized by a highly heterogeneous karstic limestone with soil pockets developing between compact and hard limestone bedrocks. The soils, that never exceed one meter depth, range from shallow calcaric Leptosol to deeper calcaric Cambisols (IUSS Working Group WRB, 2014). The litter overlying the A-horizons (O-horizons) is one to seven cm strong. The A-horizons of 2-10 cm-depth, are clayley, calcareous and show high contents in organic carbon (Table 1). These horizons have a crumbly to fine and strong subangular blocky structure likely due to a high earthworm burrowing activity and numerous fine roots. The humus is an "active oligomull or dysmull type" (Brêthes et al., 1995). The A/C horizon consists of thin layers of a clayey and fine blocky soil material between limestone rocks of a decametric size. Roots are observed inside the thin soil layers.

Downy oak (*Quercus pubescens*) and Montpellier maple (*Acer monspessulanum* L.) represent 75% and 25 %, respectively, of the foliar biomass of the dominant tree species (Kalogridis et al., 2014). The coppice, typically constituted by multiple stems sprouting from the same rooting system, is about 70 years old. Mean trees height is 5 m, and mean diameter at breast height is 10 cm, ranging from 0.9 to 18.6 cm. European smokebush (*Cotinus coggygria* Scop.) and many thermophilic and xerophilic herbaceous and grass species compose the understorey vegetation (Kalogridis et al., 2014). A network of soil sensors beneath and above the canopy, continuously records environmental parameters including: global radiation, air and soil temperature profiles, air and soil moisture, wind speed and rainfall, which are made accessible through the COOPERATE database (http://cooperate.obs-hp.fr/db).

Our understanding of the atmospheric dynamics over the O3HP sampling site does not rely solely on meteorological parameters recorded at ground level by basic weather stations. The transport and dispersion of air pollutants in the southeastern part of France was extensively investigated during the "Expérience sur Site pour COntraindre les Modèles de Pollution atmosphérique et de Transport d'Emissions" (ESCOMPTE) experiment which took place in June-July 2001 (Cros et al., 2003; Kalthoff et al., 2005). As shown by these authors for June 2001 and in Fig. S1 for June of 2012, 2013 and 2015, the sea breeze is a general characteristic of the atmospheric dynamics at the site in June. It flows from the W-SW in the afternoon and carries with it the photosmog of the city of Marseille. During the night and early morning hours the wind is orientated from other directions with a strong N-NE component (Fig. S1). However, one fundamental aspect of air circulation over the area is the existence of a nocturnal jet flowing at 800-1000 m of altitude, also with a strong N-NE

component, observed in the sodar (vertical wind profiler) measurements performed by Kalthoff et al. (2005). This is of crucial importance for the interpretation of our results.

## 2.2 Air sampling and analytical methods

### 2.2.1 Momentum, energy and $CO_2$ and isoprene fluxes

In June 2012, momentum, energy and $CO_2$ fluxes were measured at the O3HP site by the eddy covariance method using a Gill-R3-HS ultrasonic anemometer placed above the forest on a 10 m mast and a close-path infrared $CO_2$ and $H_2O$ gas analyser (IRGA, Licor 7000) placed in a truck at about 35 m from the base of the mast (Kalogridis et al., 2014). Air was drawn from an inlet located ~20 cm away from the anemometer, with a 45 m long heated PFA Teflon tubing (1/2'' OD, 3/8'' ID, heated ~1°C above ambient air temperature), at a flow rate of ~64 L min$^{-1}$, in order to maintain a turbulent flow. Air was
then sub-sampled in a tube (1/4" OD, 1/8" ID) to the close path IRGA. Data were sampled at 20 Hz. Basically, the turbulent flux of $CO_2$ was estimated as the covariance $\overline{w'c'}$ of the vertical component of the wind velocity ($w$) and the dry mole fraction of $CO_2$ ($c$), multiplied by the dry air molar volume. Here the primes denote a deviation from the mean. The friction velocity $u_* = -\sqrt{\overline{w'u'}}$, where $u$ is the along-wind air velocity component. High frequency losses corrections were estimated with the method of Ammann et al. (2006), and averaged 10% (median). The fluxes (NEE, GPP and Reco) were calculated
using the eddy covariance method as explained in Aubinet et al. (2000) and Loubet et al. (2011). In short, GPP and Reco were estimated with the method described by Kowalski et al. (2004). Briefly, the net flux of $CO_2$ (NEE) was modelled as the sum of the ecosystem respiration (Reco) and the GPP (or assimilation) modelled as a hyperbolic function of the incoming solar radiation (Rs).

NEE=-Reco+(a1·Rs)/(a2+Rs)=-Reco+GPP                                  (Eq. 3)

By convention here Reco and GPP are positive, and NEE is counted positive when carbon is fixed by the canopy. The parameters Reco, a1 and a2 were estimated by minimizing the difference between the modelled and measured $CO_2$ flux from May 16 to June 17 of 2012 using the non-linear solver in Excel and the objective function ln(mean square error between model and measurements). The comparison was only performed for well-established turbulence (u* > 0.1 m s$^{-1}$ and |z / L| < 0.2, where L is the Obukhov length), during dry periods without rain and during daytime (Rs > 5 W m$^{-2}$). The GPP was then
calculated as GPP= (a1·Rs)/(a2+Rs) for all conditions.

Q. *pubescens* is a high isoprene emitter and studies at the O3HP have shown that it is the main volatile organic compound (VOC) released by this species at the branch (Genard-Zielinski et al., 2015) and canopy scale (Kalogridis et al., 2014). Isoprene is synthetized within the leave through metabolic processes and its emission in the atmosphere is mainly controlled by temperature and radiation (Laothawornkitkul et al., 2009 and references therein). Although it does not share common
source and sink with OCS, it was used here as an additional information to understand biological processes occurring at the O3HP forest.

## 2.2.2 Carbonyl sulfide (OCS)

At the O3HP site, in June 2012, air was drawn either from an inlet located at 10 m height, ~20 cm away from the anemometer, or from a second inlet located at 2 m height on the same mast, with 70-80 m long Synflex tubing (3/8'' OD) flushed permanently at a flow rate of ~6 L min$^{-1}$. In June 2013, air was drawn solely from an inlet located at 2 m height, with 20 m long Synflex tubing (3/8'' OD). The analytical instruments were run in laboratory-like conditions (air conditioning at 25°C) in a small building away from the sampling plot. The way air was analyzed for OCS was described extensively in Belviso et al. (2013). However, the mass spectrometer detector was replaced in April 2012 by a pulsed flame photometric detector (PFPD). In general, air measurements (500 ml STP of air trapped cryogenically at 100 ml min$^{-1}$ flow rate with an ENTECH preconcentrator) were carried out on an hourly basis. Peak integration was done using the SRI's PeakSimple Chromatography Data System. Calibration was performed as in Belviso et al. (2013) but the primary standard, drawn with a gas-tight syringe, was injected in a line flushed with OCS-free helium (He was passed through an empty stainless-steel trap immersed in liquid nitrogen) connected to the preconcentror inlet. Although the calibration gas commercialized by Air Products has a tolerance of 2.5%, we found an agreement better than 0.2% between the certificate of analysis (1.013 ppm of OCS in helium) and our own measurements of that standard (1.014 ± 0.011 ppm, n=6) using a second calibration gas provided by U. Seibt and K. Maseyk who purchased it from Air Liquide (0.517 ppm in nitrogen). Since the PFPD response is quadratic, the calibration equation is obtained by plotting the natural logarithm of the peak area against the natural logarithm of OCS (picolitre or pL). Mixing ratios are calculated by dividing pL of OCS by volumes of air dried at -25°C, corrected to room temperature and pressure. Semi-continuous measurement repeatability is 1% (1 SD, n= 38 consecutive hourly analyses of atmospheric air from a compressed cylinder (target gas) containing 573 ppt of OCS). Accuracy and long-term repeatability (LTR) were better than 2.5% as evaluated from periodic analyses of an atmospheric air standard prepared and calibrated by NOAA-ESRL containing 448.6 ppt of OCS.

In June 2013, air was analyzed continuously for OCS using a commercially available OCS, $CO_2$, $H_2O$, and CO off-axis integrated cavity output spectroscopy analyzer (Los Gatos Research, Enhanced Performance Model, California, USA). In early 2013 at the O3HP, the instrument was tested for the first time in the field. We calibrated the instrument with OCS measured by the GC (over a range of atmospheric concentrations of 439 ppt to 699 ppt inherent to the period of interest for this study). OCS data collected with a ½ Hz frequency by the spectroscopy analyzer were subsequently reduced to 5-minute averages which correspond to the sampling time of the GC. The OCS signal varied by less than ± 2 ppt (standard error) in the 5 minute time window. GC and LGR data showed a linear and strong positive correlation ($OCS_{GC}$ = 1.14 $OCS_{LGR}$ + 12.3 ppt, $R^2$=0.95, n=128). Absolute readings were regularly cross-checked with a NOAA-ESRL standard showing good stability throughout the campaign. $OCS_{LGR}$ data were essentially used to document OCS variations in between GC measurements, and were scaled to GC data using the above relationship.

## 2.2.3 Carbon dioxide ($CO_2$)

At the O3HP site, in June 2012, air was analyzed for $CO_2$ from two sampling lines (10 and 2 m height), alternatively (measurement interval duration was 30 min and data collected during the first 10 min were discarded), using a commercially available PICARRO cavity ring-down spectroscopy (CRDS) analyzer (Model G2401) placed next to the OCS gas chromatograph. In addition to $CO_2$, this instrument analyzes $CH_4$ and CO mixing ratios and applies corrections for water vapor levels. Precision and stability of the measurements performed with this instrument were investigated using the rigorous testing procedures described by Yver et al. (2015) and reported in Table 1 of that manuscript (see instrument G2401 with serial number CFKADS2022 and ICOS ID 108). For $CO_2$, similar or better results in terms of continuous measurement repeatability (CMR) and LTR were obtained in the field as compared to the factory or to the test laboratory (i.e., 0.027 ppm and 0.020 ppm), respectively (Yver et al., 2015). The CRDS analyzer was calibrated in the test laboratory following ICOS standard procedures, once before shipping and right after the one month deployment in the field.

In June 2013, air was analyzed continuously for $CO_2$ using the LGR Enhanced Performance instrument (see above). $CO_2$ measurements were not reported on a calibration scale.

### 2.2.4 Carbon monoxide (CO)

At the O3HP site, in June 2012, air was analyzed for CO using the PICARRO CRDS analyzer described above. Precision in terms of CMR and LTR measured in the field was not as good as in the factory or in the test laboratory (i.e., 6.8 ppb and 2.2 ppb), respectively (Yver et al., 2015). Data were calibrated as for $CO_2$ measurements. In June 2013, air was analyzed continuously for CO using the LGR instrument. CO measurements were not reported on a calibration scale. CO was used as a semi-quantitative tracer of combustion processes (biomass or fossil fuel burning).

### 2.2.5 Ozone ($O_3$), $O_3$ deposition velocity ($V_dO_3$) and its partitioning

Ozone was measured at O3HP in June 2012 with an instrument based on ultra-violet absorption (model T-400 from API-Teledyne, San Diego, USA). This instrument, calibrated with an internal ozone generator (IZS, API) is operated with a flowrate of about 700 mL $min^{-1}$ and delivers data every minute. In June 2013, ozone concentrations measured at a few hundred meters from the main O3HP site were downloaded from the regional Air quality network Air-Paca, France, (http://www.airpaca.org/). Ozone deposition velocity ($V_dO_3$) was measured at the O3HP in June 2012 with a fast $O_3$ chemiluminescent analyser (ATDD, NOAA, USA). The Ratio Method described in Muller et al. (2010) was applied to evaluate $V_dO_3$. Detailed description of the methodology is given in Stella et al. (2011). The canopy conductance ($g_cO_3$), and non-stomatal conductance for ozone ($g_{ns}O_3$) were estimated following Lamaud et al. (2009), as $g_cO_3 = V_dO_3 / (1 - V_dO_3/V_{max}O_3)$, and $g_{ns}O_3 = g_cO_3 - g_sO_3$, where the stomatal conductance for $O_3$ ($g_sO_3$) is equal to $g_sH_2O \times 0.653$, this factor being the ratio of molecular diffusivities of $O_3$ to $H_2O$. $V_{max}O_3$ is the maximum deposition velocity for ozone which corresponds to a perfect sink of ozone at the leaf level which is the inverse of the sum of aerodynamic ($R_a$) and canopy

boundary layer resistances ($R_{bl}O_3$) as $V_{max}O_3 = 1 / (R_a + R_{bl}O_3)$, those being estimated as in Lamaud et al. (2009), taken from Bassin et al. (2004).

## 2.2.6 Stomatal conductance

Canopy stomatal conductance for water vapor ($g_sH_2O$) was estimated in 2012 from the latent (LE) and sensible (H) heat flux
from the Penman Monteith method for relative humidity $\leq$ 70 %. Under wet conditions the stomatal conductance was estimate following Lamaud et al. (2009) based on the proportionality between the assimilation of $CO_2$ and the conductance. Leaf stomatal conductance was measured in June 2013 with a porometer (AP4, Delta-T Devices, Burwell UK). Due to the unilateral distribution of stomata (hypostomatous leaf) only the abaxial sides of the leaf were measured using the 'slotted' configuration of the chamber. Five leaves were sampled per tree and cycle. Light was measured holding the sensor
horizontally above the leaf.

## 3 Results

### 3.1 Meteorological conditions and soil climate

The cumulated precipitations before the campaigns were about 400 mm and 500 mm since the beginning of the year, respectively (Fig. 1a). As few precipitation events of small intensity took place during the campaigns, the volumetric soil
water content (measured at 5 cm depth) was in a decreasing phase from about 0.3 $m^3$ $m^{-3}$ during the wet season to about 0.1 $m^3$ $m^{-3}$ during the dry season (Fig. 1b). Soil temperatures went the opposite way (Fig. 1b) and were in the range 14-19°C and 14-17°C during the 2012 and 2013 campaigns, respectively (Fig. 1c,d).

### 3.2 Diel variations in the canopy (2m)

In June of 2012, $CO_2$ presented a clear and reproducible diurnal cycle with a maximum during the night (Fig. 2c). This
maximum, an increase of 10-20 ppm, is correlated with the decrease of global radiation (Fig. 2a). This increase occurred between the period of maximum atmospheric turbulence ($u_* > 0.4$ m s$^{-1}$, Fig. 2b), a few hours after the maximum solar radiation (Fig. 2a), and the nocturnal period when atmospheric turbulence is reduced ($u_* < 0.2$ m s$^{-1}$, Fig. 2b) and strong temperature gradients above ground level form (~ -0.5 °C m$^{-1}$, Fig. 2a). The temperature gradient is a proxy of low atmospheric mixing and boundary layer stability. During this period, the variability in OCS was relatively low as compared
to $CO_2$ (10 ppt at the most). The strongest temperature gradients above ground level (~ -1°C m$^{-1}$, Fig. 2a) were observed after sunrise (4 am UTC), for about two hours. The diel cycle in the atmospheric boundary layer exhibited a much steeper decline in OCS after sunrise than during the night (Fig. 2c); the same holds for ozone (Fig. 2d). The amplitude of the early morning drop of OCS was in the 60-100 ppt range. That of $O_3$ was in the range of 15-30 ppb. It is worth to note that the large

nocturnal maximum of $CO_2$ was followed by a secondary one in the early morning, yet of shorter duration and smaller amplitude (10 ppm at the most, Fig. 2c). Hence, important variations in $CO_2$ were observed during the period of lowest OCS concentrations. In general, OCS and $O_3$ diel variations were in phase except in the late afternoon where we never observed a peak of OCS associated with the peaks of $O_3$ and CO (Fig. 3a & Fig. 2d).

Figure 4 compares the mean diel patterns in ambient OCS mixing ratios at 2m height in June 2012 and June 2013, constructed from data presented in Fig. 2c and Fig. 3b, respectively. Data show that the OCS concentrations were more stable during the night than during the day since a drop of ~ 50 ppt was observed in the early morning hours, down to ~ 450 ppt, followed by a rise up to ~ 520 ppt in June 2012 and ~ 650 ppt in June 2013. These huge diurnal variations, with amplitudes in the range of 150-250 ppt (Fig. 3b), were confirmed by independent measurements carried out with the LGR

$CO_2$/OCS/CO/$H_2O$ analyzer which was running in parallel (Fig. 3b). The concomitant decrease of OCS and $O_3$ in the early morning hours was confirmed in the 2013 records (Fig. 3b). Further, the richest air masses in $O_3$, which were transported over O3HP by strong winds in the late afternoon, were not the richest in OCS throughout the campaign (Fig. 3b). Our ground-based meteorological and ozone observations dated June of 2012 and 2013 (650 MSL) presented in Fig. 2 and Fig. 3 are highly consistent with data reported by Kalthoff et al. (2005).

**3.3 Vertical gradients**

Diel variations in near-surface OCS and $CO_2$ vertical gradients were documented twice in June 2012 from data collected alternatingly at 2 m and 10 m (Fig. 5). Both time series show no apparent OCS gradient during the night whereas $CO_2$ data showed strong vertical gradients with $CO_2$ at 2 m being higher by approximately 5 ppm than at 10 m. During the day, the $CO_2$ gradient reversed, $CO_2$ mixing ratios being lower at 2 m than at 10 m, with a back-reversal of the $CO_2$ gradient

occurring in the late afternoon at 17:00-18:00 UTC. During the day, OCS mixing ratios were systematically lower at 2 m than at 10 m by a few ppt in the morning and up to 10-20 ppt in the afternoon. Hence, $CO_2$ and OCS were consistently lower at 2 m than at 10 m during the day, during the night however, $CO_2$ had a gradient in line with the respiratory production of $CO_2$, whereas OCS showed no measurable gradient.

**3.4 Diel variations of fluxes and deposition velocities**

It should be noted here that the $CO_2$ and water fluxes are not strictly linked at the ecosystem level because the non-foliar contribution is different for $CO_2$ (non-green plant biomass, and soil respiration) and $H_2O$ (evaporation from soil and tree surfaces). Further, the gas-exchange between the sub-stomatal cavity and the atmosphere has drivers that impact differently on biological and physical processes (e.g. the temperature effect on photosynthesis and respiration for $CO_2$ and transpiration for water). However, it is known that soil water content will impact on the litter decomposition processes, and other

microbial and rooting activity that determine soil respiration. The presence of a non-stomatal water flux is an indication of the wetness of upper soil layers, and hence a proxy of an increased respiration rate. Negative water fluxes at dew point

temperature indicate dew formation which may cause non-stomatal fluxes due to the dissolution of gases. The latent heat and $CO_2$ fluxes (GPP and NEE) followed a clear diurnal cycle well correlated with global radiation, indicating that there was no significant water stress which would tend to lower the flux in the afternoon (Fig. 6a,b). However, the latent heat flux was significantly higher on June 13 than for later days (Fig. 6a). Higher water fluxes were also measured June 11 and 12 which were likely due to the evaporation of precipitations of low intensity (2 mm at the most) that occurred June 10, 11 and 12 as well as the water that was deposited as dew the nights of June 11 and 12 which was clearly shown by the air temperature reaching the dew point temperature and the sensible heat flux being highly negative at night (data not shown). The stomatal conductance for water vapor followed also a clear diurnal cycle (Fig. S2). Significant positive isoprene fluxes were only observed during daytime, following diel cycles with mid-day maxima ranging from 10 to 35 nmol m$^{-2}$ h$^{-1}$ (Fig. 6c redrawn from Kalogridis et al., 2014).

Unfortunately, the fast-$O_3$ sensor that was used to assess the $O_3$ deposition velocity had some sporadic down times which occurred frequently during the June 12 to 18 sampling period. During that period, the analyser only performed well during one night. Good quality data, however, were recorded continuously from May 29 to June 3 and, and from June 7 to 9 (Fig. 7). Stomatal conductance for $O_3$ (gs$O_3$) assessed with the method of Lamaud et al. (2009) followed diel cycles with mid-day maxima throughout the whole month of June 2012 in the range 6 to 8 mm s$^{-1}$ (data not shown but Fig. 7 provides an illustration for late May and the first week of June 2012 of the typical diel pattern of gs$O_3$). The shape of these diel cycles provides another indication that the canopy was never under water stress and the gs$O_3$ mostly light-driven. The ozone deposition velocity (V$_d$O$_3$) exhibited diurnal variations with, in general, larger deposition before mid-day (Fig. 7a). Since the stomatal conductance showed a much more symmetrical feature during daytime (Fig. 7b), it indicates that non-stomatal ozone deposition occurred preferentially during the morning. However, estimates of gns$O_3$ were less numerous in the afternoon than in the morning because of inconsistencies between gc$O_3$ and gs$O_3$ values noticed during the afternoons of May 29-31 and June 9, where gs$O_3$ was higher than gc$O_3$ (Fig. 7b). Nevertheless, in five cases out of six, a peak in gns$O_3$ was observed during the period between May 29 and June 3. Data show a shift in the relative importance of both pathways since from June 7 the ozone deposition in the morning in all cases was predominantly through the stomatal pathway. Unfortunately, we have no indication about ozone deposition pathways during the periods where OCS was monitored in the atmosphere. However, the shift towards higher $O_3$ deposition through the stomatal pathway during the second week of June (Fig. 7b) and the strong similarities between OCS and $O_3$ diurnal patterns in June 2012 (Fig. 3a), suggest that the non-stomatal pathway lost importance throughout the month of June.

**4. Discussion**

**4.1 Role of atmospheric dynamics on OCS exchange**

OCS diel variations presented here (Fig. 3) resemble those reported by Berkelhammer et al. (2014) at two sites of central North America where steep rises in OCS also occurred after sunrise (see their Fig. 7b and supplementary Fig. 11). The authors suggested that this morning rise was related to boundary layer dynamics when air from above, richer in OCS than the air from the nocturnal boundary layer, is entrained downwards. This is also the case at O3HP as shown in the vertical profiles of water vapor (Fig. S3). Entrainment of dry air from the nocturnal boundary layer is evidenced from the decrease in water vapor concentrations about two hours after sunrise. This decrease is generally more important at 10 m than at 2 m. However, diurnal variations with amplitudes over 200 ppt as observed at the O3HP in June 2013 were never reported before. This raises the question of the origin of air masses so rich in OCS advected over O3HP in mid-June 2013. It is highly unlikely that long-range transport of biomass burning gases and aerosols between North America and the Mediterranean region was responsible for OCS contamination because the transport of biomass burning material occurred in late June 2013 so after the end of our OCS surveys (see Fig.4 in Ancellet et al, 2016). As the $O_3$ rich air masses reaching the O3HP in the late afternoon are lagging those rich in OCS by ~ 4 hours (Fig. 3b), it is clear that the OCS and $O_3$ peaks have distinct origins. Backward trajectories at 300 m above ground level ending at 12 UTC (Stein et al., 2015), when OCS levels at the O3HP in June 2013 were over 600 ppt (Fig. 3b), show that the circulation of the air masses during 2012 and 2013 periods was at low altitude (below about 500 m a.g.l., i.e. below 1100 m a.s.l.), thus generally in the boundary layer. The back trajectories show that the air masses were in closer contact with the continent in June 2013 than in June 2012, and that the transport in June 2013 was from the N/NW so along the Rhône Valley (Fig. S4). South of the city of Lyon, the Rhône Valley is highly industrialized and it is therefore likely that the O3HP site is impacted by anthropogenic direct or indirect emissions of OCS (i.e. from the oxidation of $CS_2$ since the largest production of $CS_2$ in Western Europe is located in the Rhône Valley (Campbell et al., 2015)). Polluted air very likely propagates southwards in the upper layers within the nocturnal jet that was observed in the sodar measurements performed nearby at Cadarache (Kalthoff et al., 2005) and is entrained downwards in the morning when turbulence recovers. Moreover, we can also demonstrate that the source of OCS pollution is persistent from the same direction from data gathered in Fig. S5 which show the full June 2013 OCS record, starting from June 8, and the corresponding back trajectories. It is clear that there is no sign of pollution in OCS when air masses, advected from the Mediterranean Sea, reach the OHP site at noon, 300 m agl. Finally, Fig. S6 demonstrates that advection of pollutants from the combustion of fossil fuels (and from biomass burning, see above) is unlikely in the OHP area except during the night of June 15 where CO levels went up to 250 ppb. A CO pollution event was also recorded the next morning but data show no impact on OCS levels. In the afternoon, polluted air from the metropolitan area of Marseille is transported by the sea breeze thus leading to an increase of ozone at elevated layers above the convective boundary layer as demonstrated in Kalthoff et al. (2005)'s study of air circulation. The highest ozone concentrations above 100 ppb can be found about 50 km further downwind north and northeast of Marseille both at the mountainous areas of Luberon and above (Kalthoff et al., 2005; see Fig. 6 of that manuscript). We can therefore conclude that the photosmog of the city of Marseille is not a source of OCS.

## 4.2 Ecosystem relative uptake (ERU)

At the O3HP, OCS concentration gradients showing lower concentrations at 2 m than at 10 m were observed during daytime (Fig. 5), especially during the afternoon so when turbulent mixing was strongest (Fig. 1b). Gradients were inexistent during the night. This implies that the forest ecosystem was essentially a net sink of OCS. Measured $CO_2$ vertical gradients indicate that the forest ecosystem was a net sink of $CO_2$ during daytime and a net source during the night, features that were confirmed by the eddy covariance data showing NEE to range between -15 and -20 µmol m$^{-2}$ s$^{-1}$ around midday and 0-5 µmol m$^{-2}$ s$^{-1}$ during the night (Fig. 6). However, the sharp rise in OCS concentrations between 6 am and 12 am UTC (Fig. 2) and the reversal of the $CO_2$ gradients at 5-6 pm UTC (Fig. 5) reduce the time window to few hours in the afternoon where the ecosystem relative uptake of OCS (ERU), which is the ratio of the relative vertical gradients of OCS and $CO_2$, can be assessed. ERU is an important parameter since it is proportional to GPP/NEE scaled by the ratio of relative leaf exchange rates (LRU) following Eq. 2. Therefore, we anticipate that this approach to partition measured NEE will hardly be applicable at O3HP not only because the amplitude of the diurnal variations in LRU is unknown at O3HP, but also because vertical gradients of OCS cannot be calculated from measurements carried out throughout the whole period of illumination. In 2012, only data collected in the afternoon were exploitable and the mean OCS-to-$CO_2$ ratio at 2 m height was $1.33 \pm 0.02$ ppt/ppm, n=27. In June 2013, polluted air masses produced dramatic variation in atmospheric OCS-to-$CO_2$ ratios in the morning and the afternoon, leaving no time window for ERU calculation. These air masses were not related with urban photosmog episodes since there was a gap of ~ 4 hours between the peaks of OCS (up to 700 ppt) and $O_3$ (up to 85 ppb). With these caveats in mind, the ratio of the mean relative vertical gradients of OCS and $CO_2$ (calculated from linear OCS profiles) was equal to 4.7 and 4.3 for the afternoons of June 6 and 17 with, however, large relative error ($\geq 50\%$), and was consistent with ERUs reported by Blonquist et al. (2011) at the Harvard Forest AmeriFlux site in summer-autumn 2006 ($5.7 \pm 1.2$ (1 SD) for short-term ERU values calculated from linear OCS profiles as we did at the O3HP).

Only when the plant uptake is the dominant flux, the ERU is proportional to the ratio of GPP/NEE with a proportionality constant that is the LRU (Campbell et al., 2008). As discussed above, this is only the case at the O3HP site for a few hours in the afternoon (because at other moments the ecosystem is not the main driver but rather the boundary layer dynamics) and that ERU could only be calculated using the OCS and $CO_2$ gradients for these few hours. When ERUs and the mean NEE/GPP ratio calculated for the period 12-17 UTC ($0.78 \pm 0.05$, n=20) are used in Eq. 2, LRUs at the O3HP are equal to 3.7 and 3.4. These values fall in the upper range of LRUs obtained from leaf chamber studies over a large range of light conditions and tree species (1-4, Stimler et al., 2010; 1.3-2.3, Berkelhammer et al., 2014).

## 4.3 Relative role of plants and soil on OCS exchange

Our OCS measurements were carried out during the period of maximum gross primary productivity of Mediterranean oak forests (Allard et al., 2008; Maselli et al., 2014). At the O3HP, the maximum of Q. *pubescens* net photosynthetic assimilation also occurs in June (Genard-Zielinski et al., in prep). The O3HP site appears to be ideal for the use of OCS uptake by plant

as a tracer for GPP in a Mediterranean oak forest because the soil is neither a source nor a sink of OCS when GPP fluxes culminate. The lack of net uptake of OCS during the night is a specific feature to the O3HP site that is not shared by other open oak woodlands characterized by a Mediterranean climate (Kuhn et al., 1999; Sun et al., 2015). The study of Kuhn et al. (1999) was performed in June 1994 at the Hastings Natural History Reservation in Monterey County, central coastal California (490 m a.s.l.) which is located in a side valley of the Carmel Valley approximately 40 km from the coast. These authors reported a nocturnal drop in the OCS ambient mixing ratio by about 150 ppt corresponding to a nocturnal OCS deposition rate of up to -7.6 pmol $m^{-2}$ $s^{-1}$ which was estimated by a nocturnal boundary layer depletion model. The range of fluxes reported by Kuhn et al. (1999) are consistent with those measured using soil chambers at Stunt Ranch in Southern California in April 2013 (0.1 – -6.5 pmol $m^{-2}$ $s^{-1}$ ; Sun et al., 2015). OCS fluxes at Stunt Ranch exhibited clear diurnal variations with higher uptakes during the night than during the day (Sun et al., 2015). Unfortunately, the signature of these fluxes in the nocturnal boundary layer in terms of nocturnal drop in OCS mixing ratio where not reported in that manuscript. To give an illustration of what might be the atmospheric signature during stable nocturnal conditions of OCS uptake events of such intensity, we extracted data from a set of observations where the role that soil, leaf and atmospheric dynamics have on surface OCS exchange is investigated from OCS diurnal cycles (as at O3HP) and nocturnal fluxes calculated using the Radon-Tracer Method (Belviso et al., 2013). Figure S7 shows an eight day time series of ambient mixing ratios of OCS, $CO_2$, CO and $O_3$ carried out in mid-April 2015 (after bud break and almost complete leaf expansion) in a suburban area of the Saclay Plateau (Paris region), in relation to incoming global radiation, thermal stratification and wind speed (as at the O3HP). Periods of low atmospheric turbulence over the Saclay Plateau were evaluated using $^{222}$Rn accumulations. In April 2015, hourly variations show night-time and early morning decreases of OCS mixing ratios (Fig. S7c) and corresponding $^{222}$Rn increases (Fig. S7b). The amplitude of OCS diurnal variations is in the 40-80 ppt range. OCS minima coincide with calm meteorological conditions with wind velocities lower than 6 km $h^{-1}$ (Fig. S7b) which are favorable to thermal stratification (Fig. S7a), with $CO_2$ maxima sometime up to ~ 480 ppm (Fig. S7c) and with $O_3$ minima down to few ppb (Fig. S7d). However, it is worth noting here that the amplitudes of $CO_2$ and $O_3$ nocturnal variations over the Saclay Plateau in early spring are higher than those at O3HP due to anthropogenic emissions of $CO_2$, which can be traced using CO mixing ratios (Fig. S7d), and to NOx emissions which accelerate the chemical removal of $O_3$ ($O_3$ reacts with NO, data not shown). OCS fluxes calculated using the Radon-Tracer Method during stable nocturnal conditions ranged from -4.8 pmol $m^{-2}$ $s^{-1}$ (night of the 14th) to -14.2 pmol $m^{-2}$ $s^{-1}$ (night of the 11th, Fig. S7c). They fall in the upper range of fluxes reported by Kuhn et al. (1999) and Sun et al. (2015) but the comparison should be made with caution because three different methods were used to estimate the OCS fluxes (i.e., a boundary layer model, soil chambers and the Radon-Tracer Method). Qualitatively, it is clear that uptake rates of several pmol $m^{-2}$ $s^{-1}$ lead to drops in the OCS ambient mixing ratio by several tens of ppt during periods of low atmospheric turbulence. Hence, a major difference between these woodlands and the O3HP site during springtime is that soil of the Mediterranean forest ecosystem of Southern France is not a net sink of OCS. Soil OCS uptake has been shown to be dependent on soil physical properties like soil structure, water content, water-filled pore space and temperature (Van Diest and Kesselmeier, 2008; Ogée et al., 2016) but also on soil biological properties like microbial

activity (Kato et al, 2008; Ogawa et al., 2013), active roots density (Maseyk et al., 2014) or the presence of a litter layer (Berkelhammer et al., 2014; Sun et al., 2015). Away from a range of optimum uptake, which varies between soils, changes in soil water content and temperature can markedly reduce OCS uptake by soils (Van Diest and Kesselmeier, 2008). However, the soil temperature and water content at the O3HP (Fig. 1c,d) are typically in the range of optimum uptake

published by Van Diest and Kesselmeier (2008). A limitation of OCS uptake by soils due to a poor OCS diffusion is moreover unlikely considering that the soils from the O3HP are strongly structured and are far from being water saturated. Finally, the only physical property of soil differing among the three open oak woodlands is the soil texture with a fine clayey texture at the O3HP but a coarse sandy loam texture at Hastings Reservation (Kuhn et al., 1999) and at stunt Ranch (Sun et al., 2015). OCS uptake by fine-textured soils have already been reported (Maseyk et al., 2014), this result pointed out the

need for measurements of OCS uptake for a greater diversity of soils. Concerning the biological soil properties, the soil at the O3HP is covered by a relatively thick litter layer that may induce a change from OCS uptake to OCS emission (Berkelhammer et al., 2014). Sun et al. (2015) however measured at Stunt Ranch that the litter was responsible for OCS uptake. The surface horizons at the O3HP showed organic carbon contents ranging from 167 to 43 g.kg$^{-1}$ in the surface soil horizons (Table 1) but only 24 g.kg$^{-1}$ at Hastings Reservation (no data on soil organic carbon are available for Stunt Ranch).

Being richer in organic carbon, soils at the O3HP show very likely higher microbial activity, factor which should stimulate uptake of OCS by soils but apparently do not. If the capacity of soils to consume OCS is more related to specific enzymatic activities (carbonic anhydrases (CA) and OCS hydrolases) than to the general variables presented above, our observations would highlight deficiencies in these enzymatic activities in calcium carbonate rich soils of O3HP. However, this hypothesis is not consistent with the suggestion that CA performs essential role for microbial organisms to survive periods of osmotic

stress such as drought at the surface of Mediterranean soils (Wingate et al., 2008). Finally, as roots and associated rhizosphere have been found to produce OCS, a greater abundance of roots in the surface soils at O3HP by comparison to the two other oak woodlands may explain why the soils at O3HP are not a sink of OCS. In other words, the lack of nocturnal net uptake of OCS would indicate that gross consumption of this gas in soil is compensated by emission processes that remain to be characterized. No data on roots abundance are however available at Hastings Reservation or Stunt Ranch to confirm such

hypothesis.

**4.4 Potential use of OCS to partition ozone decay near the ground**

Data show strong similarities during the night and early morning hours between OCS and $O_3$ diel variations at the O3HP suggesting a similar sink during that period (Fig. 3). At the O3HP, volatile organic compounds (VOCs) produced by the vegetation are essentially in the form of isoprene (Kalogridis et al., 2014; Genard-Zielinski et al., 2014). Isoprene is

oxidized in the atmosphere by the hydroxyl radical (OH), $O_3$ and the nitrate radical ($NO_3$), but in-canopy chemical oxidation of isoprene at the O3HP was found to be weak and did not seem to have a significant impact on isoprene concentrations and fluxes above the canopy (Kalogridis et al., 2014). Hence, ozone deposition at the O3HP was essentially

through leaf uptake via stomata and surface deposition, without a strong contribution from chemical reactions.  In late May and early June 2012, the non-stomatal contribution to the ozone flux was in general markedly higher than the stomatal one in the morning hours (before 10:00 UTC) but became much less significant in the afternoon (Fig. 7b). During the second week of June, however, although there were still signs of non-stomatal loss of ozone in the morning, the major contribution to ozone deposition was through the stomatal pathway (Fig. 7b). The analogy with OCS during nighttime and early morning suggests that soil did not contribute much to the $O_3$ flux and that the deposition flux of $O_3$ in mid-June was essentially the result of leaf uptake. It is however difficult to evaluate the soil ozone pathways without turbulence measurements inside the canopy. It would be worth looking further on how OCS could be used to partition ozone fluxes near the ground between soil and leaf deposition processes. The applicability of OCS to characterize the strength of ozone sinks would be reduced in situations where NOx would significantly impact on the chemical production or destruction of ozone in the canopy or when background air is contaminated by primary or secondary anthropogenic sources of OCS (Fig. 3b).

## 5. Conclusions and perspectives

Diel changes in OCS mixing ratio and in its vertical distribution show that net soil exchange of OCS is negligible compared to the uptake of this gas through the stomata, a feature which is not shared by other oak woodland ecosystems characterized by a Mediterranean climate. Hence, O3HP would be the adequate place to support the installation in the Mediterranean region of a monitoring station of OCS uptake by plants from eddy covariance measurements. However, the assessment of GPP from measured OCS fluxes at the ecosystem scale remains tributary of our poor knowledge of LRU diel variations at the O3HP which requires further examination using new experimental facilities (branch chambers or bags and/or coupled NEE/ERU measurements). In the framework of the European infrastructure Integrated Carbon Observation System (ICOS), an atmospheric measurement station (100 m high tower) has been set up at OHP in the year 2014 to determine multi-year records of greenhouse gases. Future research on the ERU is encouraged by the site being suitable to perform continuous and high precision vertical profiles of OCS using quantum cascade laser spectrometry. Unfortunately, our preliminary surveys suggest that the site is less adequate for scaling NEE to GPP from observations of vertical gradients of OCS relative to $CO_2$ during daytime than for estimating GPP directly from eddy covariance measurements;  the time window for calculation of the ecosystem relative uptake of OCS was found to be restricted at the O3HP to few hours after midday (1) because the vertical distribution of OCS is disrupted by entrainment in the morning of OCS rich tropospheric air sometimes contaminated by anthropogenic emissions, and (2) because the $CO_2$ vertical gradient reverses when it is still light.

*Acknowledgements.*

We are grateful for the support by the administrative and technical staff of the "Observatoire de Haute-Provence" and the "Institut Mediterranéen de Biodiversité et Ecologie terrestre et marine", and support by the OHP infrastructure. We are also

grateful to Eric Lamaud and Jean-Marc Bonnefond from INRA for lending the NOAA ozone analyzer and the Li7500 CO2/H2O IRGA. The authors express their thanks to the staff of the SIRTA observatory which provided access to meteorological data. The authors gratefully acknowledge the NOAA Air Resources Laboratory (ARL) for the provision of the HYSPLIT transport and dispersion model and/or READY website (http://www.ready.noaa.gov) used in this publication.

This work was supported by the French National Agency for Research (ANR 2010 JCJC 603 01 CANOPÉE). We also thank EU FP7 ECLAIRE project for funding. The purchase of the LGR OCS, $CO_2$, $H_2O$, and CO analyzer used during the 2013 field campaign was co-funded by PACA region, GIS IBiSA, CEA, CNRS and FR 3098 ECCOREV (IMAPLANT project to B.G.).

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

Table 1: Soil physico-chemical characteristics at O3HP

| Horizon | Depth (cm) | < 2µm (g kg$^{-1}$) | 2 - 50 µm (g kg$^{-1}$) | 50–2000 µm (g kg$^{-1}$) | TOC[*] (g kg$^{-1}$) | N (g kg$^{-1}$) | pH | CaCO$_3$ (g kg$^{-1}$) |
|---------|------------|---------------------|-------------------------|--------------------------|----------------------|-----------------|-----|------------------------|
| *Leptosol* | | | | | | | | |
| A$_1$ | 0 – 5 | 560 | 340 | 96 | 167 | 8.9 | 7.1 | 6 |
| A$_2$ | 5 – 20 | 536 | 338 | 118 | 43.1 | 2.7 | 7.6 | 10.7 |
| A/C | 20 – 50 | 515 | 324 | 133 | 23.3 | 1.7 | 8.0 | 27.2 |

5  [*] Total Organic Carbon

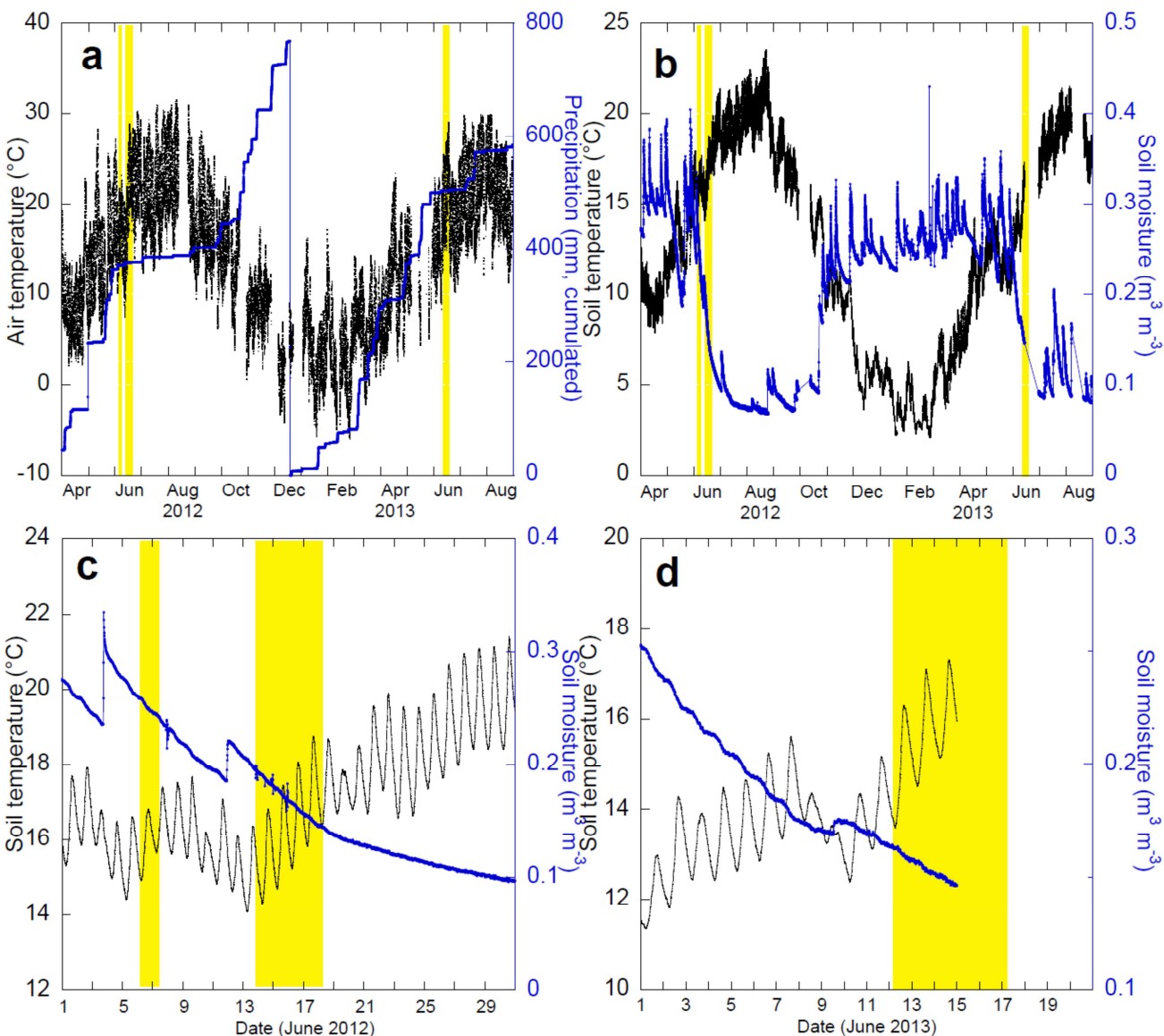

Figure 1: Monthly variations (a) in air temperature and cumulated precipitations and (b) in soil temperature and moisture (-10 cm) at an oak forest ecosystem in Southern France (O3HP). Panels c and d, same as panel b but for June 2012 and June 2013. The yellow vertical bands correspond to the sampling periods.

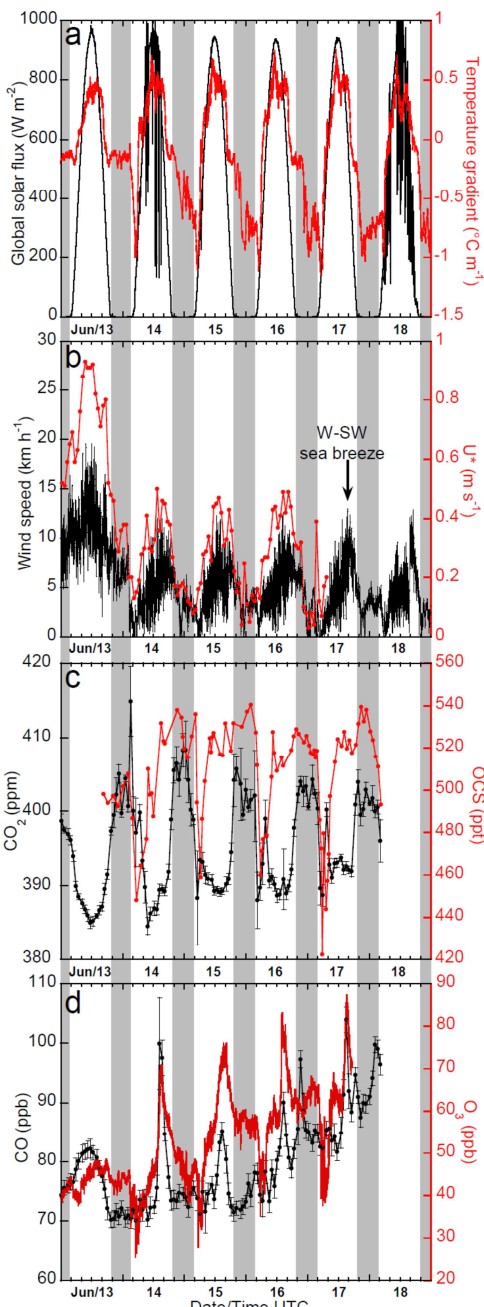

Figure 2: Time series of ambient mixing ratios of OCS, $CO_2$, CO and $O_3$ at an oak forest ecosystem in Southern France (O3HP, June 2012; c,d) at 2 m above ground level, with incoming global radiation and thermal stratification above ground level ($\Delta T/\Delta H$ in °C m$^{-1}$; a) and wind speed (b). Periods of low atmospheric turbulence were evaluated using friction velocities (u* < 0.15 m s$^{-1}$, b).The time scale is UTC time and the grey vertical bands correspond to the night time.

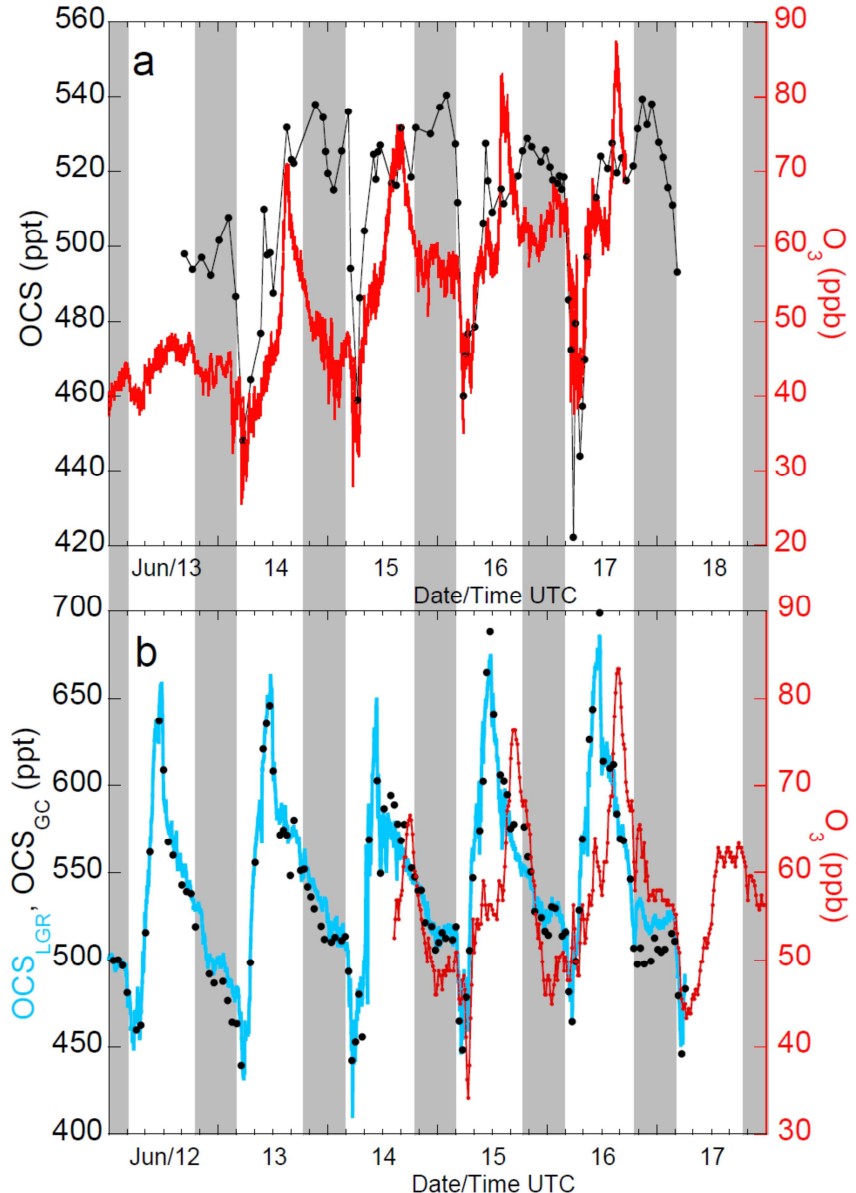

Figure 3: Diel variations in OCS and $O_3$ mixing ratios at O3HP in June 2012 (a) and June 2013 (b). In June 2013, two OCS analyzers were run in parallel and $O_3$ was measured at a few hundred meters from the main O3HP site. The LGR analyzer was calibrated against the GC, $OCS_{LGRcal.} = 1.14 \times OCS_{LGRraw} + 12.3$ ppt. $O_3$ data were downloaded from the regional Air quality network Air-Paca, France (http://www.airpaca.org/).

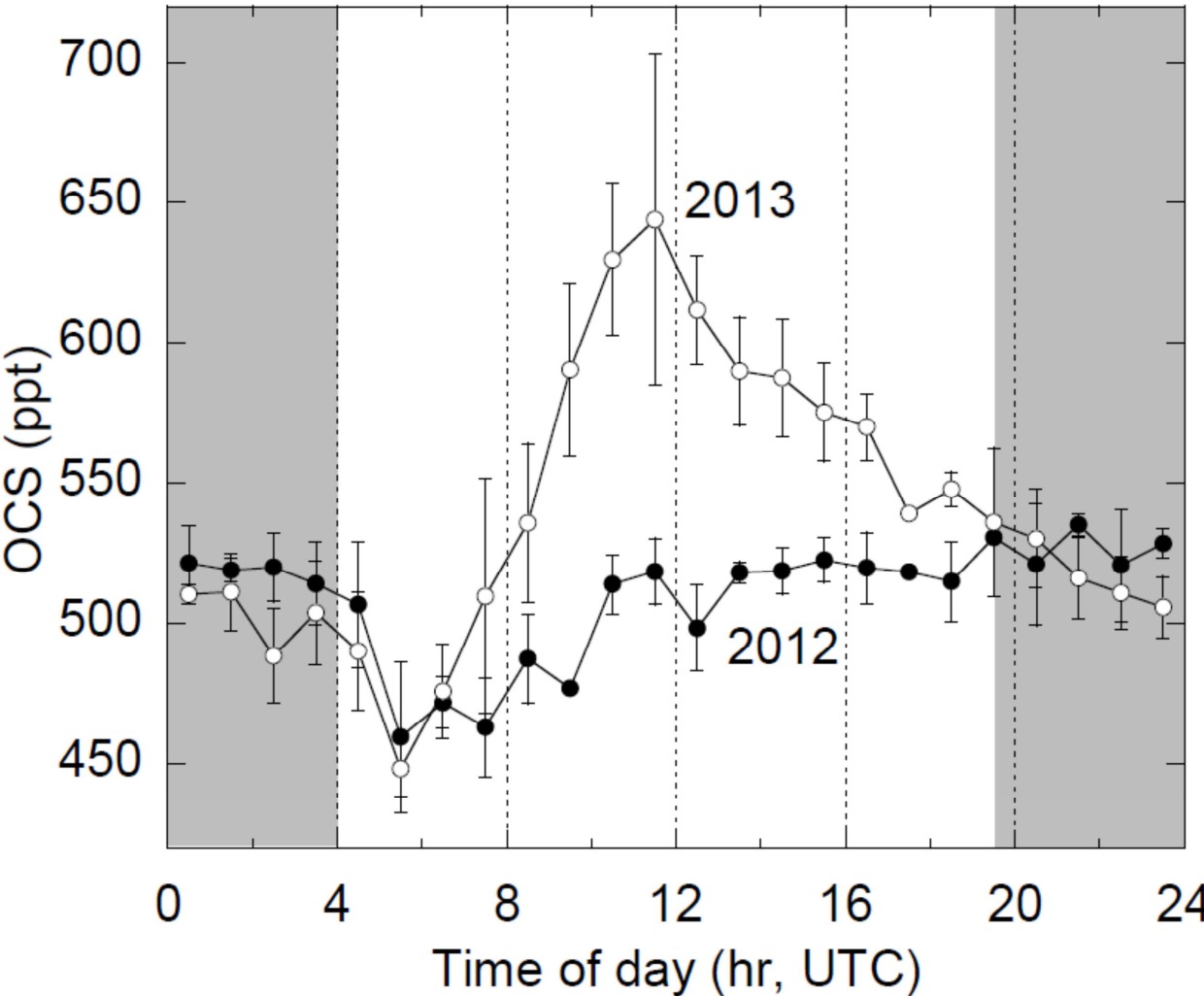

Figure 4: Mean diel patterns in ambient OCS mixing ratios at the O3HP site in June of 2012 and 2013 (displayed with dots and circles, respectively). The time scale is UTC time and the grey vertical bands correspond to the night time. Error bars represent one standard deviation of hourly mean OCS mixing ratios recorded consecutively by the gas chromatograph for several days. Full records are displayed in Fig. 2c and Fig. 3b, respectively.

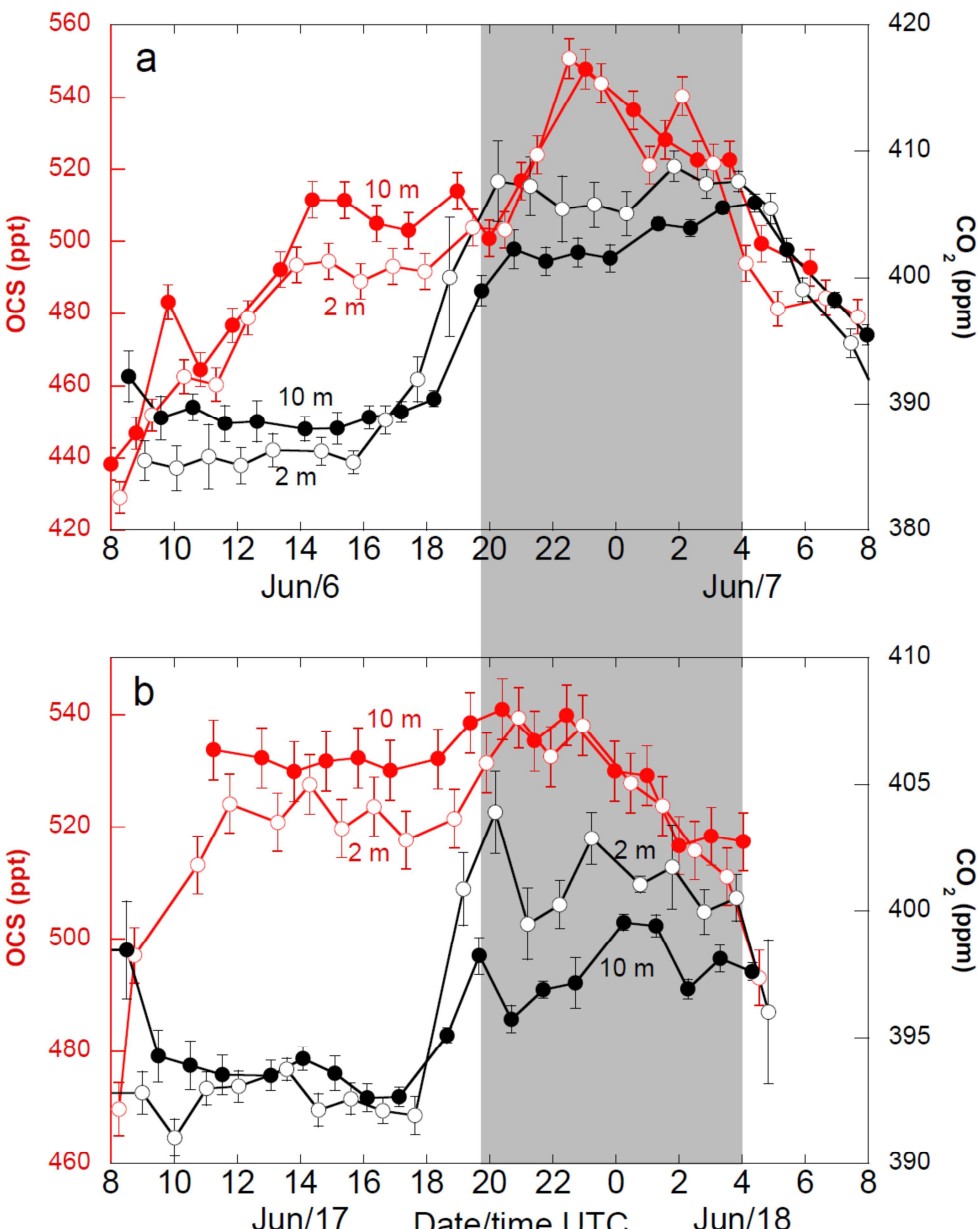

Figure 5: Time series plots showing diurnal variations in ambient OCS and $CO_2$ mixing ratios (displayed in red and black, respectively) within and above the canopy (2 m and 10 m heights, circles and dots, respectively) at the O3HP site during two measurement periods in June 2012 (a,b). The grey vertical band corresponds to the night time. Error bars represent one standard deviation of mean $CO_2$ mixing ratios recorded by the PICARRO instrument which alternated measurements between 2 m and 10 m heights on a half-hourly basis. OCS measurement repeatability is 1%.

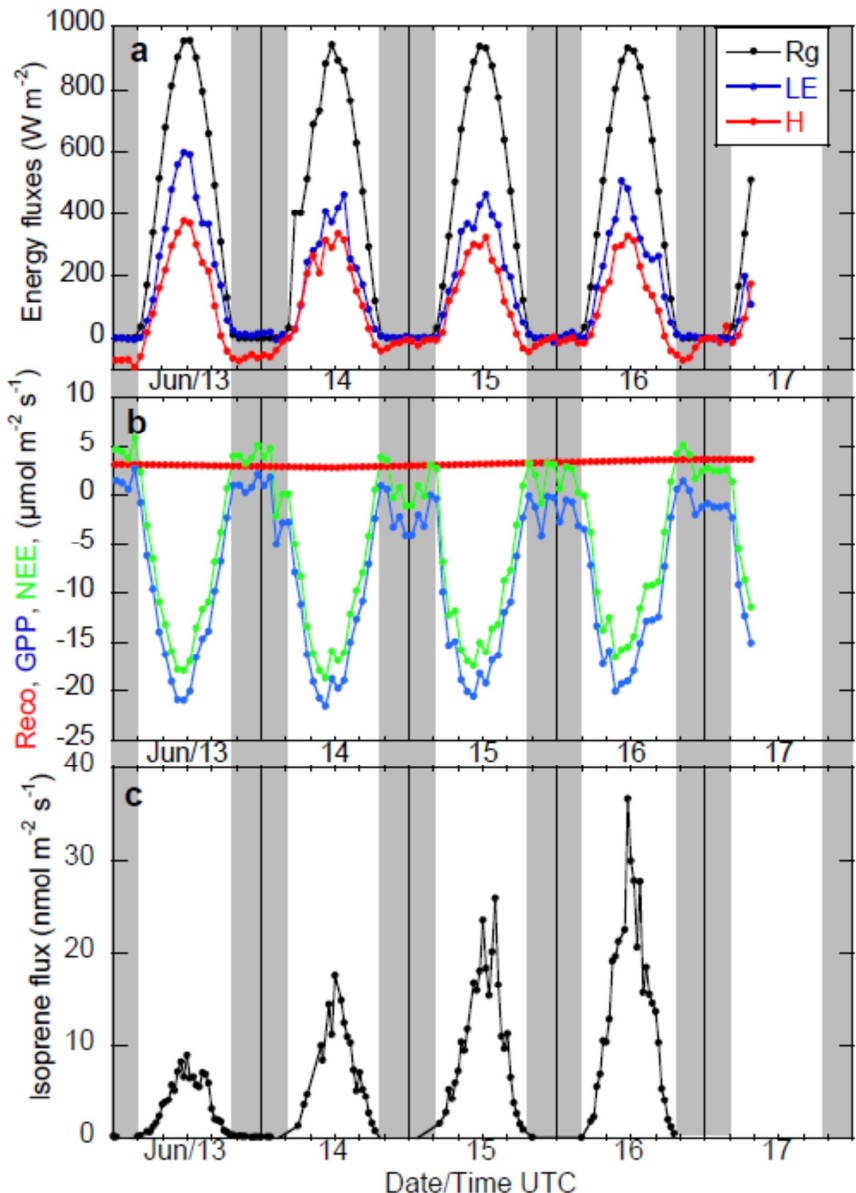

Figure 6: 4 day time series of (a) global radiation (Rg), sensible and latent heat (H and LE) and of $CO_2$ hourly fluxes from eddy covariance data measured at the O3HP site (b, June 2012). Reco, GPP and NEE fluxes stand for ecosystem respiration, gross primary production and net ecosystem exchange, respectively. We use the convention that negative values of fluxes indicate carbon uptake by the forest ecosystem. Panel (c) displays the isoprene fluxes measured concomitantly by the disjunct eddy covariance technique (Kalogridis et al., 2014).

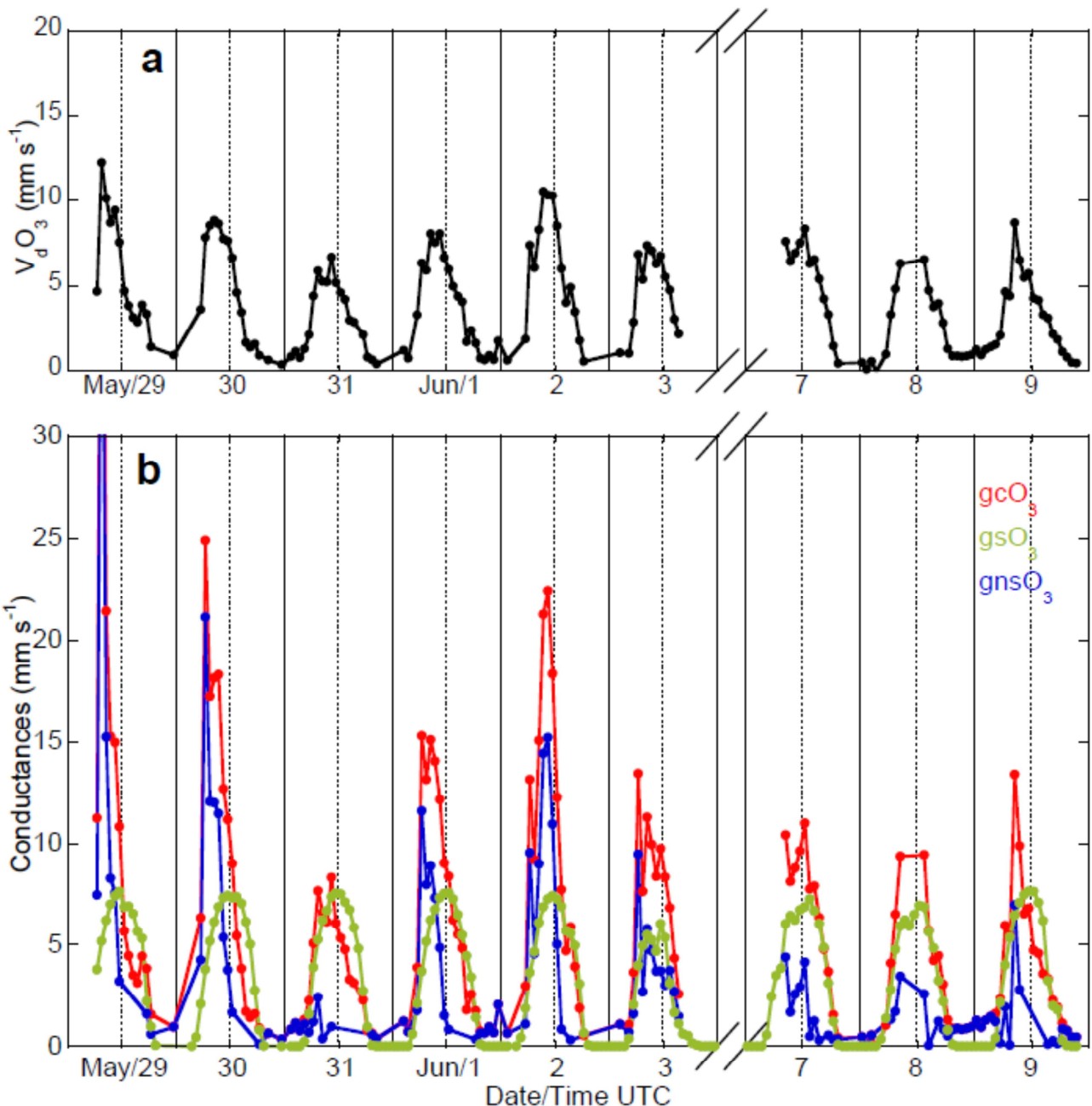

Figure 7. Diel variations in (a) ozone deposition velocity ($V_dO_3$) and (b) canopy conductance ($gcO_3$), stomatal conductance ($gsO_3$) and non-stomatal conductance ($gnsO_3$) over the May 29 to June 9 period in 2012. The partitioning was obtained with the Lamaud et al. (2009) approach (see text for details).

