# Peer review of "A top-down approach of surface carbonyl sulfide exchange by a Mediterranean oak forest ecosystem in Southern France"

_Atmospheric Chemistry and Physics, 2016_

## Referee Comment (RC1) · Anonymous Referee #2 · 8 Jul 2016

Overview

I read this manuscript on carbonyl sulfide with great interest because the data provide several advances over related studies including continuous observations, multiple tracers, and multiple sample heights. I agree with the conclusions in this study but have several comments that may clarify the role of this field site within the broader context of carbon cycle science. In particular, the variation in LRU needs to be better framed to note that this variation is critical for canopy-scale studies but it is less critical than other uncertainties for regional-scale studies.

Specific Comments

1) "However, there is evidence that the Leaf Relative Uptake of OCS and of CO2 (LRU), ... following Eq. (1) (Campbell et al., 2008; Asaf et al., 2013),..." The introduction begins with a discussion of LRU variability that should be revised to clarify a key distinction with respect to spatial scales. In particular, the impact of LRU variability for the regional-scale analysis in Campbell et al. (2008) is very different from the impact of LRU variability on canopy-scale analysis. At the regional scale, the effect of LRU variability is less significant because the regional spatial uncertainty in GPP is much larger than the LRU uncertainty. This is demonstrated in Hilton et al. (Tellus B, 2015) by showing that the mechanistic simulation of COS plant flux by SiB and a constant LRU implementation of COS plant flux in SiB have small differences in comparison to the large differences between multiple ecosystem models (SiB, CASA, CLM, etc.). However, at the canopy-scale the temporal variation in LRU becomes much more significant when trying to infer daily or even hourly GPP fluxes use COS observations. The reader should be reminded of this distinction again later in the paper where the authors write "However, the assessment of GPP from measured OCS fluxes remains tributary of our poor knowledge of the magnitude of the LRU diel variations which requires further examination."

Hilton, T.W., Kulkarni, S., Zumkehr A., Berry, J.A., Campbell, J.E. (2015) Large variability in ecosystem models explains a critical parameter for quantifying GPP with atmospheric carbonyl sulfide, Tellus B, v16, http://dx.doi.org/10.3402/tellusb.v67.26329.

2) The discussion could be expanded to note paths forward for addressing diel LRU variation for canopy scale analysis. For example, some field studies are now making canopy flux and leaf chamber measurements simultaneously and using leaf chambers to estaimte LRU and then using canopy measurements to estiamte GPP.

3) Further discussion of the high mixing ratios observed in 2013 could be added. These observations coincide with back trajectories to the Rhône Valley. The gridded anthropogenic inventory of Kettle et al. (2002) does not show significant emissions in this region. However the Kettle inventory used a coarse spatial proxy that is not specific

to the COS source industries. A more recent inventory developed by Campbell et al. (2015) uses industry-specific data and finds that the primary anthropogenic source is the indirect source from industrial CS2 emissions. These industry data show that Adisseo France in the Rhône Valley is the largest producer of carbon disulfide in Western European.

4) It would be interesting to expand figure 4 to plot the COS/CO2 ratio for the multiple years and sample heights. This ratio is of interest because it also appears in equation 1 and is being used by multiple modeling groups to scale GPP.

5) The authors report that their ERU of 4.3 is similar to Harvard Forest values. They may also want to expand the comparison to note similarities to more spatially diverse data including a range of 2-8 reported for North American NOAA airborne data (Montzka et al., 2007) and 5.7 +- 0.6 reported for North American NASA airborne data (Campbell et al., 2008).

6) The soils were not a net sink which contrasted with field measurements from Sun et al. which show a soil sink in the Stunt Ranch oak field site. However, recent laboratory incubations using soil samples from Stunt Ranch have found that Stunt Ranch soils could also result in no net sink (or even a small net source) under certain temperature and soil moisture conditions (Whelan et al., 2016).

Whelan M.E., Hilton T.W., Berry J.A., Berkelhammer M., Desai A.R., Campbell J.E. (2016) Carbonyl sulfide exchange in soils for better estimates of ecosystem carbon uptake. Atmos. Chem. Phys. 16, 3711-3726, doi:10.5194/acp-16-3711-2016..

7) The night/day shading in many of the figures is a great visual cue and could be added to Figure 3 also.
* * *

---

## Referee Comment (RC2) · Anonymous Referee #1 · 28 Jul 2016

Carbonyl sulfide has been postulated a while ago as a potential proxy that may be used to estimate gross primary production at flux tower sites. In the present manuscript Belviso et al. present measurements of the diurnal dynamics of OCS, $CO_2$, and $O_3$ and of their respective fluxes above a Mediterranean oak forest ecosystem during two summer campaigns. The authors analyze the applicability of the OCS-GGP approach, and the suitability of the site as a flux monitoring station in a Mediterranean climate. Based on their data they elaborate and discuss the problems and limitations of their concept in an open and thorough manner. A major problem of the site apparently arises from the advection of pollution-derived OCS that occasionally flaws OCS gradients towards the vegetation sink. The manuscript as a whole is crafted very well. I support

the conclusions drawn. These interesting new data clearly deserve publication in ACP. From multiple readings of the manuscript I cannot find reasons why it should not be published in almost its present form. Below please find some minor remarks.

Minor remarks

I would like to encourage citation of the publication that first reported stomatal uptake of OCS molecule by leaves (the central mechanism of the paper), not only of the most recent publications on page 1. To my knowledge this has first been published by Paul Goldan (Goldan et al., Journ. Geophys. Res., 93, 14186-14192, 1988).

I recommend to consider moving the introduction of the second approach to estimate GPP (including equation (2)) from chapter 4.2 to the introduction in chapter 1, next to equation (1). This way the conceptual frame of using OCS as a tracer gets clearer, and it does not come as a "surprise" in chpt. 4.2.

P.7, L. 18: ". . . the range in OCS was relatively low . . .", do you mean "the variability" ?

---

## Referee Comment (RC3) · Anonymous Referee #3 · 15 Sep 2016

General comments:

The authors present atmospheric OCS concentrations during a few days in June 2012 and June 2013 and tried to explain the variability by considering the processes in soil, vegetation, and atmospheric transport. Large changes in atmospheric OCS are observed at the site, including large decreases in the early morning, and large increases in the afternoon for data from 2013. I think the authors can do further analysis to show that the reasons that they give for the increasing OCS concentrations are indeed plausible.

The lack of an afternoon peak in 2012 is explained by the fact that for these days the backward trajectories show that the air was transported mostly from the South, and not

from the industrialized area in the Rhône Valley. I suggest the authors consider doing a windrose analysis to show if the source of OCS is persistent from the same direction.

The authors suggest that the early-morning drop in OCS is caused by vegetative uptake and that it increases shortly after that due to entrainment of air from above the boundary layer. The authors could try to make the existence of entrainment more plausible by looking at for example H2O concentrations. The air above the atmospheric boundary layer is generally drier than within the boundary layer. If the increased OCS concentrations are indeed driven by entrainment, then also a decrease in water vapor concentrations can be expected.

Another dynamical process that should be considered is the sea breeze. Due to large convection over land there is generally lower pressure over land, which causes air to move from sea to land during daytime. The authors suggest that the high peak of ozone in the afternoon data in 2013 is transported by the sea breeze with the source in the Marseille area. I wonder why the ozone peak should come from the Marseille area, and not from the Rhône valley. The wind direction should be shown to indicate the presence of a sea breeze and the correlation with the ozone peak. Furthermore, if the enriched air of OCS is coming from an industrial area a correlation with CO would be expected, was this visible at the site? An analysis using wind direction and other tracers (e.g. H2O for entrainment, CO for advection from industrialized areas) must be done to better characterize the sources (and sinks) of OCS.

In general I wonder why the authors only show data from a few days in June 2012 and June 2013. Did they only measure during these few days? Please point out if these were only two short campaigns. If the authors have a longer measurement period available they should explain why they chose to show only a few days and I suggest they consider including a longer time series of data. This would have added value in characterizing the atmospheric dynamics and the sources and sinks of OCS at the site. For example, by considering a longer time series of data the authors can discuss if the afternoon peaks observed in June 2013 are actually a rare event or if they occur more

often. Besides that, they can discuss if sea breezes are a general characteristic of the atmospheric dynamics at the site. The currently presented measurement period is rather short to draw conclusions on the suitability of the site to study OCS as a tracer for GPP. In fact, the current data show that the ecosystem OCS uptake is not a dominant process for most of the day (e.g. influence of entrainment in the morning and pollution in the afternoon). The authors showed that ERU calculations were limited to only a few hours, which actually suggests that this site is not ideal to study OCS as a tracer for GPP.

Specific comments:

Abstract: Page 1, line 27: I would think it is relevant to say from which absolute concentrations the values drop. E.g. say "... and synchronous steep drops of OCS from ... ppt down to ... ppt". The same holds for O3.

Introduction: Page 2, line 17: "Atmospheric OCS is also removed from the atmosphere by other pathways, such as nighttime uptake by plants...". I would not use the word "pathway" here, as the nighttime uptake by plants follows the same pathway as the daytime uptake by plants, that is, through open stomata. Only the difference with CO2 is that the OCS uptake is not light-dependent, and therefore it is not corresponding with photosynthesis.

Material and Methods: Section 2.1. Site description: It would be worth mentioning the canopy height.

Page 4, lines 16-21. Can the authors briefly explain the method to partition GPP and Reco?

Page 5, lines 1-9: What is exactly a calibration gas provided by U. Seibt and K. Maseyk who purchased it from Air Liquide? $\sim$1 ppm or $\sim$500 ppt OCS? How did the authors find an agreement better than 0.2%?

Page 5, lines 10-11: What was compared/evaluated? Was the cylinder air from NOAA-

ERRL used as target? Replace "certified" with "calibrated".

Page 5. Did the authors observe a dependence with water vapor when comparing OCSLGR with OCSGC? Kooijmans et al., (2016) found that for the Aerodyne laser spectrometers there can be spectral interference between H2O and OCS, depending on the spectral fit. Did the authors observe something similar?

Page 5, line 28: I think "this manuscript" refers to Yver et al. (2015)? In that case I suggest the authors say "that manuscript".

Page 6, line 24-27. Both methods seem to be used under wet conditions: "Penman Monteith for RH > 70 %" and "Under wet conditions the stomatal conductance was estimated following Lamaud et al. (2009)". Did the authors mean to say that one of the two methods is used under dry conditions?

Results: Page 7, line 6: "The two campaigns took place in June of 2012 and 2013." I suggest the authors mention this earlier in the manuscript, e.g. in the introduction or in the methods. This would make clear already in the methods section that some instrumentation for one variable differs over the two years. Besides that, please explain why only the data from a few days in 2012 and 2013 were used and not a longer time series.

Page 7, line 21: "... same for ozone". Better say: "... and the same holds for ozone."

Page 8, Line 17-21: Is there any relation between the increased water flux and CO2 fluxes? What does this information tell us? I do not see a further discussion about the latent heat fluxes in the discussion session, so does this information have added value?

Page 8, line 21: Maybe the authors can introduce already before what the relation is between isoprene fluxes and CO2 fluxes. That would make clear why the authors measure this. Discussion:

Page 9, line 12: As explained in the general comments I suggest the authors look at

[Figure]

H2O concentrations to see if the morning rise of OCS coincides with a decrease in H2O concentrations, which may be an indication of entrainment.

Page 9, line 17-20: Please explain this better, was there a typical event of excessive biomass burning in North America that could potentially have explained the OCS increase?

Page 9, line 21: "... it is clear that the OCS and O3 peaks have distinct origins". The air has the same origin, but the OCS and O3 enrichment has different sources.

Page 9, line 23: "Backward trajectories at 300 m above ground level ending at 12 UTC, when OCS levels at the O3HP in June 2013 were over 600 ppt, show that the circulation of air masses during both periods was at low altitude...". Define "both periods". Do the authors mean 2012 and 2013? The sentence before points to only 2013 data.

Page 9, line 24-30: It is not clear here what message the authors try to convey. The authors point out two different trajectories: one is from the Rhône Valley, where anthropogenic emissions could cause a rise in OCS. The other is the sea breeze, which (I presume) could transport the high O3 concentrations from the Marseille area, but this peak does not coincide with the OCS peak.

Page 9, line 28-29: The authors state that polluted air from the Marseille area is transported by a sea breeze, leading to an increase of ozone above the boundary layer. Why would a sea breeze cause transport above the boundary layer? I would say this transport happens within the boundary layer as a sea breeze causes horizontal movement from the sea towards land. Please also show why it is plausible that there is a sea breeze, did the wind direction change? Why would the Marseille area cause an ozone peak and not the Rhône Valley? And did ozone correlate with CO for the 2013 data?

Page 10, line 10: the authors probably mean to refer to Fig. 5 instead of 3.

Page 10, line 14: ERU is defined as the ratio of the relative drawdown of OCS to CO2.

Only when the plant uptake is the dominant flux, the ERU is proportional to the ratio of GPP/NEE with a proportionality constant that is the LRU (Campbell et al., 2008). Please make clear that the formulation that the authors use is only valid when the plant uptake is the dominant flux. After that the authors can explain that this is only the case at the OH3P site for a few hours in the afternoon (because at other moments the ecosystem is not the main driver but rather the boundary layer dynamics) and that ERU could only be calculated using the OCS and CO2 gradients for these few hours. Please give the numbers reported by Blonquist et al. (2011). I am also aware of ERU values presented by Maseyk et al. (2014). What do these ERU values tell us about the plant uptake? (Like the authors state in the third reason given in the beginning of section 4.3, see also my next comment).

Page 10, line 26-28: please clarify all three reasons to reach the conclusion that OCS uptake is the only relevant biospheric flux. This is not clear yet.

Page 11, line 31-33: Please rephrase, it reads as if the authors refer to the difference between the three open oak woodlands. But the authors probably mean the difference between these woodlands and the O3HP site. Also be more explicit how this conclusion is obtained: "The fact that no large nighttime drop of OCS is observed at O3HP suggests that the soil is not a net sink of OCS." The soil temperature and moisture have not changed from 2012 to 2013, and a early morning drawdown was indeed observed in 2012.

Page 12, line 10. Remove "If"

Conclusions and perspectives: Page 13, line 15. Which requirements? Introduce them in the introduction and repeat here. Did the authors refer to the spring in 2012 only?

Page 13, line 15-17: The authors state that the soil uptake of OCS is negligible compared to the uptake of this gas through the stomata, however, I think this conclusion is made too easily. In fact no net exchange of OCS during the night is observed, which could either mean that there is no soil and leaf flux during the night, or that the sources

and sinks (either from the soil or leaves) compensate each other. State clearly that this is just a speculation.

Page 13, line 21: which "second method" do the authors mean? Which is the first?

Page 13, line 19-22: The authors discuss here that LRU is needed to derive GPP from OCS fluxes, and then continue saying that there were difficulties in determining ERU. To my knowledge LRU can only be derived from leaf-level measurements with branch chamber/bag measurements (e.g. Berkelhammer et al., 2014), how do your perspectives tackle the issue of getting LRU values?

Figures

Fig 2. 2012 data are shown, but the 2013 data are at least as important due to the high afternoon OCS peaks. I suggest the authors show both the 2012 and 2013 data. Also interesting to see would be the wind direction as an indication for a sea breeze and H2O as indication for entrainment.

Fig 3a. This can already be seen from Fig 2c and 2d. I suggest the authors include meteo and concentration data of 2013 in Fig 2, then remove Fig 3, and include the average daily cycle of ozone in Fig 4 (to still be able to make the comparison between OCS and ozone).

Fig 5. Please show uncertainty bars for OCS as for CO2.

---

## Author Comment (AC2) · 26 Oct 2016

Please consult the attached document.

Please also note the supplement to this comment:
http://www.atmos-chem-phys-discuss.net/acp-2016-525/acp-2016-525-AC2-supplement.pdf

---

## Author Comment (AC4) · 26 Oct 2016

*Dear Editor,*

*On behalf of co-authors of the manuscript entitled "A top-down approach of surface carbonyl sulfide exchange by a Mediterranean oak forest ecosystem in Southern France", I would like to thank reviewer 1 and reviewer 2 for stating that they agree with the conclusions drawn in this manuscript. Reviewer 3 asked us to do further analysis to show that the reasons that we give for the increasing OCS concentrations are indeed plausible. We agree that we need to provide a better description of the atmospheric dynamics for the region during late spring-early summer which is, according to the proceedings of the ESCOMPTE experiment (Cros et al., 2004; Kalthoff et al. 2005), in fact rather well understood. Our meteorological observations and those of the ESCOMPTE experiment are in good agreement, i.e. that major changes in wind directions occur in the area throughout the day and that the photosmog of the city of Marseille (high in $O_3$) is advected to the sampling site by the afternoon sea breeze. In this response to the reviewers' comments and in the supplementary information, we now provide a series of figures from which it is apparent that the source of OCS responsible for the huge increase of OCS is located inland (Fig. S5). This OCS is unlikely to be related to combustion processes (Fig. S6) but more to indirect production of OCS from industrial $CS_2$ emissions in the Rhône Valley where the largest production of $CS_2$ in Western Europe is located. Observations by sodar measurements performed at Cadarache (Kalthoff et al., 2005) show that the polluted air most likely propagates southwards in the upper layers within the nocturnal jet and is entrained downwards by morning turbulence. Evidence of air entrainment is provided from the vertical profiles of water vapor shown in Fig. S3. The photosmog of the city of Marseille is high in $O_3$ but not in OCS.*

*The other comments were all addressed.*

*References:*
*Cros, B., P. Durand, H. Cachier, Ph. Drobinski, E. Fréjafon, C. Kottmeier, P.E. Perros, V.-H. Peuch, J.-L. Ponche, D. Robin, F. Saïd, G. Toupance, and H. Wortham. The ESCOMPTE program: an overview. Atmospheric Research 69: 241–279, 2004.*

*Kalthoff, N., C. Kottmeier, J. Thürauf, U. Corsmeier, F. Saïd, E. Fréjafon, and P.E. Perros. Mesoscale circulation systems and ozone concentrations during ESCOMPTE: a case study from IOP 2b. Atmospheric Research 74: 355–380, 2005.*

The new documents are organized as follows:
**-Reviewers' comments**
*-Our responses to the reviewers's comments*
-3 figures reproduced from other articles (Kalthoff et al., 2015; Ancellet et al., 2016)
-4 new figures available in the supplementary material (Fig. S1, S3, S5, S6)
-Corrections in the revised manuscript

**Anonymous Referee #1**

Carbonyl sulfide has been postulated a while ago as a potential proxy that may be used to estimate gross primary production at flux tower sites. In the present manuscript Belviso et al. present measurements of the diurnal dynamics of OCS, CO2, and O3 and of their respective fluxes above a Mediterranean oak forest ecosystem during two summer campaigns. The authors analyze the applicability of the OCS-GGP approach, and the suitability of the site as a flux monitoring station in a Mediterranean climate.

Based on their data they elaborate and discuss the problems and limitations of their concept in an open and thorough manner. A major problem of the site apparently arises from the advection of pollution-derived OCS that occasionally flaws OCS gradients towards the vegetation sink. The manuscript as a whole is crafted very well. I support the conclusions drawn. These interesting new data clearly deserve publication in ACP.

From multiple readings of the manuscript I cannot find reasons why it should not be published in almost its present form. Below please find some minor remarks.

*Thank you very much.*

**Minor remarks**

**I would like to encourage citation of the publication that first reported stomatal uptake of OCS molecule by leaves (the central mechanism of the paper), not only of the most recent publications on page 1. To my knowledge this has first been published by Paul Goldan (Goldan et al., Journ. Geophys. Res., 93, 14186-14192, 1988).**

*You are absolutely right. We added the following two sentences in the introduction:*
*"In the late 80's, vegetation has been proposed to be the missing sink in the global cycle of atmospheric carbonyl sulfide (OCS; Brown and Bell, 1986; Goldan et al., 1988) and the first evidence from field observations of the uptake of OCS near the ground was provided by Mihalopoulos et al. (1989)."*
*"The global network NOAA ESRL for measurements of greenhouse gases in the atmosphere monitors OCS mixing ratios on a weekly basis since year 2000 (Montzka et al., 2007). It is in this framework that the major role of vegetation in the global budget of OCS was again emphasized."*

**I recommend to consider moving the introduction of the second approach to estimate GPP (including equation (2)) from chapter 4.2 to the introduction in chapter 1, next to equation (1). This way the conceptual frame of using OCS as a tracer gets clearer, and it does not come as a "surprise" in chpt. 4.2.**

*The following sentence was added at the end of the introduction:*
*"Since direct LRU and OCS flux measurements were not performed during the campaigns, we used the ecosystem relative uptake (ERU) approach of Campbell et al. (2008) to provide a rough estimation of LRU variations using the following equation:*

$$LRU = [ERU].[NEE / GPP] \qquad (2)$$

*where ERU is the relative gradient of OCS ($m^{-1}$) divided by the relative gradient of $CO_2$ ($m^{-1}$) and NEE is the net ecosystem exchange of $CO_2$ from eddy covariance measurements carried out at the site."*

**P.7, L. 18: "...the range in OCS was relatively low...", do you mean "the variability" ?**
*Correct.*

**Anonymous Referee #2**

**Overview**
**I read this manuscript on carbonyl sulfide with great interest because the data provide several advances over related studies including continuous observations, multiple tracers, and multiple sample heights. I agree with the conclusions in this study but have several comments that may clarify the role of this field site within the broader context of carbon cycle science. In particular, the variation in LRU needs to be better framed to note that this variation is critical for canopy-scale studies but it is less critical than other uncertainties for regional-scale studies.**

*Thank you very much.*

**Specific Comments**
**1) "However, there is evidence that the Leaf Relative Uptake of OCS and of CO2 (LRU),... following Eq. (1) (Campbell et al., 2008; Asaf et al., 2013),..." The introduction begins with a discussion of LRU variability that should be revised to clarify a key distinction with respect to spatial scales. In particular, the impact of LRU variability for the regional-scale analysis in Campbell et al. (2008) is very different from the impact of LRU variability on canopy-scale analysis. At the regional scale, the effect of LRU variability is less significant because the regional spatial uncertainty in GPP is much larger than the LRU uncertainty. This is demonstrated in Hilton et al. (Tellus B, 2015) by showing that the mechanistic simulation of COS plant flux by SiB and a constant LRU implementation of COS plant flux in SiB have small differences in comparison to the large differences between multiple ecosystem models (SiB, CASA, CLM, etc.). However, at the canopy-scale the temporal variation in LRU becomes much more significant when trying to infer daily or even hourly GPP fluxes use COS observations. The reader should be reminded of this distinction again later in the paper where the authors write "However, the assessment of GPP from measured OCS fluxes remains tributary of our poor knowledge of the magnitude of the LRU diel variations which requires further examination."**
**Hilton, T.W., Kulkarni, S., Zumkehr A., Berry, J.A., Campbell, J.E. (2015) Large variability in ecosystem models explains a critical parameter for quantifying GPP with atmospheric carbonyl sulfide, Tellus B, v16, http://dx.doi.org/10.3402/tellusb.v67.26329.**

*The introduction was partly rewritten following your recommendations. It now reads:*
*"In the late 80's, vegetation has been proposed to be the missing sink in the global cycle of atmospheric carbonyl sulfide (OCS; Brown and Bell, 1986; Goldan et al., 1988) and the first evidence from field observations of the uptake of OCS near the ground was provided by Mihalopoulos et al. (1989). Nowadays, the mechanistic link between leaf $CO_2$ and OCS exchange is well understood (Stimler et al., 2010; Seibt et al., 2010; Wohlfahrt et al., 2012) and the scientific community has reached consensus on the potential of atmospheric OCS measurements to provide independent constraints on GPP at canopy (Blonquist et al., 2011; Asaf et al., 2013), regional (Campbell et al., 2008) and global (Montzka et al., 2007; Berry et al., 2013; Launois et al., 2015) scales. However, recent studies also demonstrated limitations to the use of OCS as a GPP proxy at canopy and ecosystem scales because (1) consumption and/or production of OCS occur in soil and litter (Van Diest and Kesselmeier 2008; Sun et al., 2015; Ogée et al., 2016; Whelan et al., 2016 and references therein), (2) in agricultural fields and midlatitude forests OCS can be taken up by plants also by night (Maseyk et al., 2014; White et al., 2010; Commane et al., 2015), and (3) the leaf relative uptake of OCS and of $CO_2$ (LRU), which is of central importance in the calculation of GPP from eddy covariance measurements of OCS exchange ($L_{OCS}$) following Eq. (1), exhibit daily and seasonal variations of variable amplitudes (Berkelhammer et al., 2014; Maseyk et al., 2014; Commane et al., 2015).*
$$GPP = (L_{OCS} / LRU).([CO2] / [OCS]) \tag{1}$$
*The character L in $L_{OCS}$ stands for leaf because OCS exchange equals $L_{OCS}$ when other ecosystem fluxes are negligible. To address the diel LRU variations and the role of soil and litter for canopy scale*

*analysis, some research groups are now combining canopy flux, leaf and soil chamber measurements in the field (L. Kooijmans personal communication, Sep. 2016).*
*Eq. 1 can also be used for regional scale analysis (Campbell et al., 2008). At this scale, LRU also varies as a function of plant type (i.e. C3 vs. C4 plants, Stimler et al., 2011). However, Hilton et al. (2015) demonstrated that the effect of LRU variability was less significant at regional than at canopy scale because the regional spatial uncertainty in GPP is much larger than the LRU uncertainty."*

*Hilton et al. (2015) was added to the reference list*

**2) The discussion could be expanded to note paths forward for addressing diel LRU variation for canopy scale analysis. For example, some field studies are now making canopy flux and leaf chamber measurements simultaneously and using leaf chambers to estimate LRU and then using canopy measurements to estimate GPP.**

*This is now stated in the revised introduction.* **"***To address the diel LRU variations and the role of soil and litter for canopy scale analysis, some research groups are now combining canopy flux, leaf and soil chamber measurements in the field (L. Kooijmans personal communication, Sep. 2016)."*

**3) Further discussion of the high mixing ratios observed in 2013 could be added. These observations coincide with back trajectories to the Rhône Valley. The gridded anthropogenic inventory of Kettle et al. (2002) does not show significant emissions in this region. However the Kettle inventory used a coarse spatial proxy that is not specific to the COS source industries. A more recent inventory developed by Campbell et al. (2015) uses industry-specific data and finds that the primary anthropogenic source is the indirect source from industrial CS2 emissions. These industry data show that Adisseo France in the Rhône Valley is the largest producer of carbon disulfide in Western European.**

*You are absolutely right. We added the following sentences:*
*"South of the city of Lyon, the Rhône Valley is highly industrialized and it is therefore likely that the O3HP site is impacted by anthropogenic direct or indirect emissions of OCS (i.e. from the oxidation of $CS_2$ since the largest production of $CS_2$ in Western Europe is located in the Rhône Valley (Campbell et al., 2015)). Polluted air very likely propagates southwards in the upper layers within the nocturnal jet that was observed in the sodar measurements performed nearby at Cadarache (Kalthoff et al., 2005) and is entrained downwards in the morning when turbulence recovers. Moreover, we can also demonstrate that the source of OCS pollution is persistent from the same direction from data gathered in Fig. S5 which show the full June 2013 OCS record, starting from June 8, and the corresponding back trajectories. It is clear that there is no sign of pollution in OCS when air masses, advected from the Mediterranean Sea, reach the OHP site at noon, 300 m agl. Finally, Fig. S6 demonstrates that advection of pollutants from the combustion of fossil fuels (and from biomass burning, see above) is unlikely in the OHP area except during the night of June 15 where CO levels went up to 250 ppb. A CO pollution event was also recorded the next morning but data show no impact on OCS levels. In the afternoon, polluted air from the metropolitan area of Marseille is transported by the sea breeze thus leading to an increase of ozone at elevated layers above the convective boundary layer as demonstrated in Kalthoff et al. (2005)'s study of air circulation. The highest ozone concentrations above 100 ppb can be found about 50 km further downwind north and northeast of Marseille both on the mountainous areas of Luberon and above (Kalthoff et al., 2005; see Fig. 6 of that manuscript). We can therefore conclude that the photosmog of the city of Marseille is not a source of OCS."*

**4) It would be interesting to expand figure 4 to plot the COS/CO2 ratio for the multiple years and sample heights. This ratio is of interest because it also appears in equation 1 and is being used by multiple modeling groups to scale GPP.**

*We calculated the OCS/CO$_2$ ratio for 2012 for the period 8h-16h UTC. A new sentence was added in text: "In 2012, only data collected in the afternoon were exploitable and the mean OCS-to-CO$_2$ ratio at 2 m height was 1.33 ± 0.02 ppt/ppm, n=27." Unfortunately the LGR instrument used in June 2013 was not calibrated. We prefer not to compare calibrated with uncalibrated data.*

**5) The authors report that their ERU of 4.3 is similar to Harvard Forest values. They may also want to expand the comparison to note similarities to more spatially diverse data including a range of 2-8 reported for North American NOAA airborne data (Montzka et al., 2007) and 5.7 +- 0.6 reported for North American NASA airborne data (Campbell et al., 2008).**

*The ERU were recalculated and the values obtained are equal to 4.7 and 4.3 for the afternoons of June 6 and 17. We prefer to compare short-term ERU measurements measured at the canopy scale with other measurements carried out in similar conditions. That is why we added the following sentence in text: "With these caveats in mind, the ratio of the mean relative vertical gradients of OCS and CO$_2$ (calculated from linear OCS profiles) was equal to 4.7 and 4.3 for the afternoons of June 6 and 17 with, however, large relative error (≥ 50%), and was consistent with ERUs reported by Blonquist et al. (2011) at the Harvard Forest AmeriFlux site in summer-autumn 2006 (5.7 ± 1.2 (1 SD) for short-term ERU values calculated from linear OCS profiles as we did at the O3HP)."*

**6) The soils were not a net sink which contrasted with field measurements from Sun et al. which show a soil sink in the Stunt Ranch oak field site. However, recent laboratory incubations using soil samples from Stunt Ranch have found that Stunt Ranch soils could also result in no net sink (or even a small net source) under certain temperature and soil moisture conditions (Whelan et al., 2016).**

*We think it is more appropriate to compare field observations with other field observations than with samples manipulated in the laboratory.*

**Whelan M.E., Hilton T.W., Berry J.A., Berkelhammer M., Desai A.R., Campbell J.E. (2016) Carbonyl sulfide exchange in soils for better estimates of ecosystem carbon uptake. Atmos. Chem. Phys. 16, 3711-3726, doi:10.5194/acp-16-3711-2016..**

**7) The night/day shading in many of the figures is a great visual cue and could be added to Figure 3 also.**
*The night/day shading was added in Fig. 3.*

**Anonymous Referee #3**

**General comments:**
**The authors present atmospheric OCS concentrations during a few days in June 2012 and June 2013 and tried to explain the variability by considering the processes in soil, vegetation, and atmospheric transport. Large changes in atmospheric OCS are observed at the site, including large decreases in the early morning, and large increases in the afternoon for data from 2013. I think the authors can do further analysis to show that the reasons that they give for the increasing OCS concentrations are indeed plausible.**

*Hope that the reviewer will support the conclusions drawn in this manuscript in the light of figures copied from other papers and the new ones we prepared.*

**The lack of an afternoon peak in 2012 is explained by the fact that for these days the backward trajectories show that the air was transported mostly from the South, and not from the industrialized area in the Rhône Valley. I suggest the authors consider doing a windrose analysis to show if the source of OCS is persistent from the same direction.**
*Our understanding of the atmospheric dynamics over the O3HP sampling site does not rely solely on meteorological parameters recorded at ground level by basic weather stations. The transport and dispersion of air pollutants in the southeastern part of France was extensively investigated during the "Expérience sur Site pour COntraindre les Modèles de Pollution atmosphérique et de Transport d'Emissions" (ESCOMPTE) experiment which took place in June-July 2001 (Cros et al., 2003; Kalthoff et al., 2005). The O3HP site is named station S21 on the figure shown below, reproduced from Kalthoff et al.'s Fig. 1.*

[Figure]

Fig. 1. Positions of the stations in the ESCOMPTE domain. The corresponding names of the stations are given in Table 1. The grey scale indicates the height of orography.

*About 25 km south of station S21 is the Cadarache sampling site (named SO1) were horizontal wind vectors and ozone concentrations were measured on June 25, 2001, by sodar and lidar, respectively. The temporal variations in wind speed, wind direction and ozone concentrations recorded by both vertical profilers were gathered in the figure shown below, reproduced from Kalthoff et al.'s Fig. 6.*

[Figure]

Fig. 6. Horizontal wind vectors measured by sodar (SO1) and ozone concentrations measured by lidar (L2) at Cadarache on June 25, 2001. The red dot indicates the ozone value measured by the Dornier 128 over the Cadarache site.

*The authors stated that "within the nocturnal boundary layer, which extends up to 1 km MSL, northeasterly to easterly winds occur, accompanied by a jet-like wind speed profile. The jet reaches about 8 m s⁻¹ at 750 m MSL…The jet can also be observed in the sodar measurements performed at Cadarache (SO1). It develops at about 2 UTC and reaches its maximum at about 6 UTC (Fig. 6)… The sodar measurements also reveal the change from the nocturnal down-valley wind system to the daytime up-valley wind system which establishes at about 9 UTC." The authors also stated that "At Cadarache in the Durance valley, about 60 km inland, the ozone maximum at the surface and at flight level 920 m MSL appears between 14 and 15 UTC… Hence, in the lower layers polluted air moves along the valley, while in the upper layers, the polluted air crosses the southern sidewalls of the Durance valley and propagates northwards. This finding highlights the height-dependent advection of ozone due to interacting mesoscale circulation systems." Since our ground-based meteorological and ozone observations dated June of 2012 and 2013 (650 MSL) presented in Fig. 2 and Fig. 3 were highly consistent with data reported by Kalthoff et al. (2005) in their Fig. 6, we stated that "In the afternoon, polluted air from the metropolitan area of Marseille is transported by the sea breeze thus leading to an increase of ozone at elevated layers above the convective boundary layer. The highest ozone concentrations above 100 ppb can be found about 50 km further downwind north and northeast of Marseille both on the mountainous areas of Luberon and above (Kalthoff et al., 2005; see Fig. 6 of that manuscript)." Hence, it is unlikely that ozone from the Rhône valley is transported to the O3HP site during the afternoon. The existence of a nocturnal jet with a strong component from the NE makes more plausible the transport above the sampling site of OCS from the Rhône and Durance upper valleys.*

*This being said, it is clear that a subsection should be added in chapter 2 (Materials and methods) to describe the atmospheric dynamics of the area established from previous surveys. The following paragraph was added:*
*"Our understanding of the atmospheric dynamics over the O3HP sampling site does not rely solely on meteorological parameters recorded at ground level by basic weather stations. The transport and dispersion of air pollutants in the southeastern part of France was extensively investigated during the "Expérience sur Site pour COntraindre les Modèles de Pollution atmosphérique et de Transport*

d'Emissions" (ESCOMPTE) experiment which took place in June-July 2001 (Cros et al., 2003; Kalthoff et al., 2005). As shown by these authors for June 2001 and in Fig. S1 for June of 2012, 2013 and 2015, the sea breeze is a general characteristic of the atmospheric dynamics at the site in June. It flows from the W-SW in the afternoon and carries with it the photosmog of the city of Marseille. During the night and early morning hours the wind is orientated from other directions with a strong N-NE component (Fig. S1). However, one fundamental aspect of air circulation over the area is the existence of a nocturnal jet flowing at 800-1000 m of altitude, also with a strong N-NE component, observed in the sodar (vertical wind profiler) measurements performed by Kalthoff et al. (2005). This is of crucial importance for the interpretation of our results."

In the results chapter, we now state that "Our ground-based meteorological and ozone observations dated June of 2012 and 2013 (650 MSL) presented in Fig. 2 and Fig. 3 are highly consistent with data reported by Kalthoff et al. (2005)." Important changes in wind direction take place in the area throughout the day. That is why a classical windrose analysis is not adequate. Instead, a plot has been prepared showing the wind direction at ground level in June of 2012, 2013 and 2015, and at 100 m height (in June 2015) recorded at the ICOS-OHP tall tower which became operational in autumn of 2014 (see new Fig. S1 reproduced below).

[Figure]

Figure S1: Analyses of wind directions recorded at ground level in June of 2012, 2013 (10 min mean) and 2015 (selected data, hourly mean), and at the top of the ICOS-OHP tall tower (100m agl, June 2015, selected data). This height is intermediate between ground level measurements (this work) and sodar vertical wind profiles (Kalthoff et al., 2015). Winds orientated from the NW-NE sector are highlighted in cyan.

*To show that the source of O₃ is persistent from the same direction, Fig. 2b was slightly modified by adding "W-SW sea breeze" in the panel at the position of maximum wind speed (not shown below). Moreover, we can also demonstrate that the source of OCS pollution is persistent from the same direction from data gathered in Fig. S5 which show the full June 2013 OCS record, starting from June 8, and the corresponding back trajectories. It is clear that there is no sign of pollution in OCS when air masses, advected from the Mediterranean Sea, reach the OHP site at noon, 300 m agl.*

[Figure]

Figure S5: Full OCS records in June 2013 and corresponding back trajectories of 48h duration to better visualize the movement of air masses over the Mediterranean Sea and France. Note that the x-axis is reversed and that there is a gap in the time series due to GC maintenance and calibration on Jun/11. The Jun/10 back trajectory shows that the air mass travelled from the Mediterranean Sea up to 45°N, then moved backwards to reach the OHP site. This may explain why the maximum OCS level on that day is not as high as later in the week.

**The authors suggest that the early-morning drop in OCS is caused by vegetative uptake and that it increases shortly after that due to entrainment of air from above the boundary layer. The authors could try to make the existence of entrainment more plausible by looking at for example H2O concentrations. The air above the atmospheric boundary layer is generally drier than within the boundary layer. If the increased OCS concentrations are indeed driven by entrainment, then also a decrease in water vapor concentrations can be expected.**

*This point is also addressed in a new figure available in the supplementary material (new Fig. S3 reproduced below).*

[Figure]

Figure S3: Time series of ambient mixing ratios of OCS (GC data) and water vapor (Picarro data) within (2 m) and above the canopy (10 m, only for water vapor). The time scale is UTC and the dashed lines indicate the sunrise.

*After sunrise, a peak in water vapor (evapotranspiration) is associated with a low in OCS (photosynthesis). Then entrainment of dry air from the nocturnal boundary layer is evidenced from the decrease in water vapor concentrations. The decrease is generally more important at 10m than at 2m. Consequently, the increased OCS concentrations in the morning are indeed driven by entrainment.*

**Another dynamical process that should be considered is the sea breeze. Due to large convection over land there is generally lower pressure over land, which causes air to move from sea to land during daytime. The authors suggest that the high peak of ozone in the afternoon data in 2013 is**

**transported by the sea breeze with the source in the Marseille area.  I wonder why the ozone peak should come from the Marseille area, and not from the Rhône valley.  The wind direction should be shown to indicate the presence of a sea breeze and the correlation with the ozone peak. Furthermore, if the enriched air of OCS is coming from an industrial area a correlation with CO would be expected, was this visible at the site?  An analysis using wind direction and other tracers (e.g. H2O for entrainment, CO for advection from industrialized areas) must be done to better characterize the sources (and sinks) of OCS.**

*This point has already been addressed above (see Fig. S5 and Fig. 2b), the analysis of $H_2O$ vertical profiles for entrainment too (see Fig. S3). The analysis using CO for advection from the city of Marseille and from industrial areas was done to characterize the sources of OCS in June 2012 (see Fig. 2d). It wasn't done for June 2013 because the LGR instrument was not calibrated. Nevertheless, the following figure now available in the supplementary material Fig. S6 demonstrates that advection of pollutants from the combustion of fossil fuels (and from biomass burning, see above) is unlikely in the OHP area except during the night of June 15 where CO levels went up to 250 ppb. A CO pollution event was also recorded the next morning but data show no impact on OCS levels.*

[Figure]

Figure S6: Time series of ambient mixing ratios of OCS (GC data) and CO (LGR uncalibrated data) within the canopy (2 m).

**In general I wonder why the authors only show data from a few days in June 2012 and June 2013. Did they only measure during these few days?  Please point out if these were  only  two  short campaigns.  If  the  authors  have  a  longer  measurement  period available they should explain why they chose to show only a few days and I suggest they consider including a longer time series**

**of data. This would have added value in characterizing the atmospheric dynamics and the sources and sinks of OCS at the site. For example, by considering a longer time series of data the authors can discuss if the afternoon peaks observed in June 2013 are actually a rare event or if they occur more often.**

*Indeed, these were only short but intensive campaigns. That of June 2013 lasted more than five days since the time series started June 8 and ended June 17. Since the GC was stopped for maintenance and calibration on day 11, we decided not to show the survey from June 8 to 10. The new Fig. S5 presents the full record and shows that 6 of 8 diurnal cycles display an OCS peak at noon UTC. This peak was not detected when the air was advected from the Mediterranean Sea as shown in Fig. S5. Hence, it is apparent that the source of OCS responsible for the huge increase of OCS is located inland (Fig. S5). This OCS is unlikely to be related to combustion processes (Fig. S6) but more to indirect production of OCS from industrial $CS_2$ emissions in the Rhône Valley where the largest production of $CS_2$ in Western Europe is located. Observations by sodar measurements performed at Cadarache (Kalthoff et al., 2005) show that the polluted air most likely propagates southwards in the upper layers within the nocturnal jet and is entrained downwards by morning turbulence. Evidence of air entrainment is provided from the vertical profiles of water vapor shown in Fig. S3. The photosmog of the city of Marseille is high in $O_3$ but not in OCS.*

*In June 2012, unfortunately, we didn't measure OCS during more than the few days we report on.*

**Besides that, they can discuss if sea breezes are a general characteristic of the atmospheric dynamics at the site.**
*According to Kalthoff et al. (2005), that's indeed the case. See the new paragraph shown above beginning with: "Our understanding of the atmospheric dynamics over the O3HP sampling site does not rely solely on meteorological parameters recorded at ground level by basic weather stations…"*

**The currently presented measurement period is rather short to draw conclusions on the suitability of the site to study OCS as a tracer for GPP. In fact, the current data show that the ecosystem OCS uptake is not a dominant process for most of the day (e.g. influence of entrainment in the morning and pollution in the afternoon). The authors showed that ERU calculations were limited to only a few hours, which actually suggests that this site is not ideal to study OCS as a tracer for GPP.**

*OCS measurements during the month of June are ideal because, according to Allard et al. (2008) and Maselli et al. (2014), the maximum gross primary productivity of Mediterranean oak forests occurs during that period. At the O3HP, the maximum of Q. pubescens net photosynthetic assimilation also occurs in June (Genard-Zielinski et al., in prep). The O3HP site is ideal to study OCS as a tracer for GPP in a Mediterranean oak forest because the soil is neither a source nor a sink of OCS when GPP fluxes culminate. This is an important advantage that should be put in the balance. We agree that the entrainment of polluted air in the morning can be a strong disadvantage but we lack of pluri-annual data to assess the frequency of such pollution events. Our observations suggest that systematic analyses of back trajectories at noon and at 300m above ground level could help for that purpose.*

**Specific comments:**
**Abstract: Page 1, line 27: I would think it is relevant to say from which absolute concentrations the values drop. E.g. say "...and synchronous steep drops of OCS from...ppt down to...ppt". The same holds for O3.**

*We think that it is more important to mention the amplitude of the variations than to mention from which absolute concentrations the values drop. That is why we modified the sequence as follows: "(amplitude in the range of 60-100 ppt) and O3 (amplitude in the range of 15-30 ppb)".*

**Introduction: Page 2, line 17: "Atmospheric OCS is also removed from the atmosphere by other pathways, such as nighttime uptake by plants...". I would not use the word "pathway" here, as the nighttime uptake by plants follows the same pathway as the daytime uptake by plants, that is, through open stomata. Only the difference with CO2 is that the OCS uptake is not light-dependent, and therefore it is not corresponding with photosynthesis.**

*The sentence has been removed and the introduction partly rewritten as follows:*
*"In the late 80's, vegetation has been proposed to be the missing sink in the global cycle of atmospheric carbonyl sulfide (OCS; Brown and Bell, 1986; Goldan et al., 1988) and the first evidence from field observations of the uptake of OCS near the ground was provided by Mihalopoulos et al. (1989). Nowadays, the mechanistic link between leaf $CO_2$ and OCS exchange is well understood (Stimler et al., 2010; Seibt et al., 2010; Wohlfahrt et al., 2012) and the scientific community has reached consensus on the potential of atmospheric OCS measurements to provide independent constraints on GPP at canopy (Blonquist et al., 2011; Asaf et al., 2013), regional (Campbell et al., 2008) and global (Montzka et al., 2007; Berry et al., 2013; Launois et al., 2015) scales. However, recent studies also demonstrated limitations to the use of OCS as a GPP proxy at canopy and ecosystem scales because (1) consumption and/or production of OCS occur in soil and litter (Van Diest and Kesselmeier 2008; Sun et al., 2015; Ogée et al., 2016; Whelan et al., 2016 and references therein), (2) in agricultural fields and midlatitude forests OCS can be taken up by plants also by night (Maseyk et al., 2014; White et al., 2010; Commane et al., 2015), and (3) the leaf relative uptake of OCS and of $CO_2$ (LRU), which is of central importance in the calculation of GPP from eddy covariance measurements of OCS exchange ($L_{OCS}$) following Eq. (1), exhibit daily and seasonal variations of variable amplitudes (Berkelhammer et al., 2014; Maseyk et al., 2014; Commane et al., 2015).*
*GPP = ($L_{OCS}$ / LRU).([$CO_2$] / [OCS])*
*(1)*
*The character L in $L_{OCS}$ stands for leaf because OCS exchange equals $L_{OCS}$ when other ecosystem fluxes are negligible. To address the diel LRU variations and the role of soil and litter for canopy scale analysis, some research groups are now combining canopy flux, leaf and soil chamber measurements in the field (L. Kooijmans personal communication, Sep. 2016).*
*Eq. 1 can also be used for regional scale analysis (Campbell et al., 2008). At this scale, LRU also varies as a function of plant type (i.e. C3 vs. C4 plants, Stimler et al., 2011). However, Hilton et al. (2015) demonstrated that the effect of LRU variability was less significant at regional than at canopy scale because the regional spatial uncertainty in GPP is much larger than the LRU uncertainty."*

**Material and Methods: Section 2.1. Site description: It would be worth mentioning the canopy height.**
*It is now mentioned in the text: "Mean trees height is 5 m, and mean diameter at breast height is 10 cm, ranging from 0.9 to 18.6 cm."*

**Page 4, lines 16-21. Can the authors briefly explain the method to partition GPP and Reco?**
*New paragraph added to methods: "The fluxes (NEE, GPP and Reco) were calculated using the eddy covariance method as explained in Aubinet et al. (2000) and Loubet et al. (2011). In short, GPP and Reco were estimated with the method described by Kowalski et al. (2004). Briefly, the net flux of $CO_2$ (NEE) was modelled as the sum of the ecosystem respiration (Reco) and the GPP (or assimilation) modelled as a hyperbolic function of the incoming solar radiation (Rs).*
*NEE=-Reco+(a1·Rs)/(a2+Rs)=-Reco+GPP                    (Eq. 3)*
*By convention here Reco and GPP are positive, and NEE is counted positive when carbon is fixed by the canopy. The parameters Reco, a1 and a2 were estimated by minimizing the difference between the modelled and measured $CO_2$ flux from May 16 to June 17 of 2012 using the non-linear solver in Excel and the objective function ln(mean square error between model and measurements). The*

comparison was only performed for well-established turbulence (u* > 0.1 m s-1 and |z / L| < 0.2, where L is the Obukhov length), during dry periods without rain and during daytime (Rs > 5 W m-2)."

The GPP was then calculated as GPP= (a1·Rs)/(a2+Rs) for all conditions. The following figure shows the adjustment between the modelled and the measured $CO_2$ flux during that period as a function of the solar radiation (Rs). It shows a rather good agreement with variability around the model and especially during night time.

[Figure]

Comparison of measured $CO_2$ fluxes with the adjusted modelled fluxes using the Kowalski et al. (2004) model. The adjusted parameters are Reco = 3.9 μmol m$^{-2}$ s$^{-1}$, a1 = 40 μmol J$^{-1}$ and a2 = 900 W m$^{-2}$.

**Page 5, lines 1-9: What is exactly a calibration gas provided by U. Seibt and K. Maseyk who purchased it from Air Liquide?~ 1 ppm or ~ 500 ppt OCS? How did the authors find an agreement better than 0.2%?**

*The grade certified working class OCS standard purchased by U. Seibt in October 2011 from Air Liquide was 0.517 ppm. The GC was calibrated with Ulli's standard. Then, we analyzed the OCS content of our own standard purchased in August 2011 from Air Products. We measured 1.014 ± 0.011 ppm (n=6), and compared this result with the certificate of analysis provided by Air Products (1,013 ± 0.025 ppm). The relative difference ((1.014 – 1.013)/1.014) is 0.1%. That is why we wrote "better than 0.2%".*

*The sentence was corrected accordingly: "Although the calibration gas commercialized by Air Products has a tolerance of 2.5%, we found an agreement better than 0.2% between the certificate of analysis (1.013 ppm of OCS in helium) and our own measurements of that standard (1.014 ± 0.011 ppm, n=6) using a second calibration gas provided by U. Seibt and K. Maseyk who purchased it from Air Liquide (0.517 ppm in nitrogen)."*

**Page 5, lines 10-11: What was compared/evaluated? Was the cylinder air from NOAA-ESRL used as target? Replace "certified" with "calibrated".**

*That's right, the cylinder air from NOAA-ESRL was used as target because our primary standard is supplied by Air Products.*

**Page 5. Did the authors observe a dependence with water vapor when comparing OCSLGR with OCSGC? Kooijmans et al., (2016) found that for the Aerodyne laser spectrometers there can be spectral interference between H2O and OCS, depending on the spectral fit. Did the authors observe something similar?**

*According to manufacturer specifications of our instrument, OCS dry fraction mixing ratio (corrected for water vapor interferences: dilution and spectroscopic effects) does not change measurably as a*

*function of water vapor mixing ratio for the range of 5000-15000 ppm. Furthermore, our simultaneous GC and LGR measurements showed very robust relationship with low variability (SD less than 10.5 ppt) between OCS measurements using dry air sampling (GC) and air of ambiant humidity (LGR) which supports such a low dependency on water vapor. The 99 % confidence interval for the empirical relationship between LGR and GC data, established in field conditions, has been found to be +- 1.9 ppt at mean ambient OCS concentrations of 520 ppt, and increases to +-4.7 ppt and +-7.4 ppt for the range of +-100 ppt and +-200 ppt, respectively, around the ambient mean (based on n=193 measurements, full $OCS_{GC}$ dataset for June 2013).*

**Page 5, line 28: I think "this manuscript" refers to Yver et al. (2015)? In that case I suggest the authors say "that manuscript".**
*Taken into account in the revised MS.*

**Page 6, line 24-27. Both methods seem to be used under wet conditions: "Penman Monteith for RH > 70 %" and "Under wet conditions the stomatal conductance was estimated following Lamaud et al. (2009)". Did the authors mean to say that one of the two methods is used under dry conditions?**
*We actually meant "Penman Monteith for RH $\leq$ 70 %" and not "> 70%". We thank the reviewer for spotting this error.*

**Results: Page 7, line 6: "The two campaigns took place in June of 2012 and 2013." I suggest the authors mention this earlier in the manuscript, e.g. in the introduction or in the methods. This would make clear already in the methods section that some instrumentation for one variable differs over the two years. Besides that, please explain why only the data from a few days in 2012 and 2013 were used and not a longer time series.**

*This is now mentioned in the methods.* "The two campaigns took place in June of 2012 and 2013. Both were of short duration (i.e. about two-weeks long)."

**Page 7, line 21: "...same for ozone". Better say: "...and the same holds for ozone."**
*Taken into account in the revised MS.*

**Page 8, Line 17-21: Is there any relation between the increased water flux and CO2 fluxes? What does this information tell us? I do not see a further discussion about the latent heat fluxes in the discussion session, so does this information have added value?**

*It should be noted here that the $CO_2$ and water fluxes are not strictly linked at the ecosystem level because the non-foliar contribution is different for $CO_2$ (non-green plant biomass, and soil respiration) and $H_2O$ (evaporation from soil and tree surfaces). Further, the gas-exchange between the sub-stomatal cavity and the atmosphere has drivers that impact differently on biological and physical processes (e.g. the temperature effect on photosynthesis and respiration for $CO_2$ and transpiration for water). However, it is known that soil water content will impact on the litter decomposition processes, and other microbial and rooting activity that determine soil respiration. The presence of a non-stomatal water flux is an indication of the wetness of upper soil layers, and hence a proxy of an increased respiration rate. Negative water fluxes at dew point temperature indicate dew formation which may cause non-stomatal fluxes due to the dissolution of gases.*

**Page 8, line 21: Maybe the authors can introduce already before what the relation is between isoprene fluxes and CO2 fluxes. That would make clear why the authors measure this.**
*Q. pubescens is a high isoprene emitter and studies at the O3HP have shown that it is the main volatile organic compound (VOC) released by this species at the branch (Genard-Zielinski et al., 2015) and canopy scale (Kalogridis et al., 2014). Isoprene is synthetized within the leave through metabolic*

processes and its emission in the atmosphere is mainly controlled by temperature and radiation (Laothawornkitkul et al., 2009 and references therein). Although it does not share common source and sink with OCS, it was used here as an additional information to understand biological processes occurring at the O3HP forest. This short paragraph is now available in chapter 2 "Materials and methods".

**Discussion: Page 9, line 12:  As explained in the general comments I suggest the authors look at H2O concentrations to see if the morning rise of OCS coincides with a decrease in H2O concentrations, which may be an indication of entrainment.**
*This point is discussed above.*

**Page 9, line 17-20:  Please explain this better, was there a typical event of excessive biomass burning in North America that could potentially have explained the OCS increase?**
*Indeed, biomass burning material was transported from North America but the events happened later in June 2013 as shown below (Ancellet et al, 2016).*

[Figure]

**Figure 4.** Map of the relative fraction of the FLEXPART biomass burning tracer plume in % for the Canadian (top) and Colorado (bottom) fires on 27 June 2013 at 06:00 UT (left) and 28 June at 18:00 UT (right). The altitude range corresponds to the vertical levels included in the calculation of the tracer relative fraction.

*Biomass burning is a source of CO but we found no sign of combustion events in the CO records of 2012 during the morning hours (Fig. S6).*

**Page 9, line 21: "...it is clear that the OCS and O3 peaks have distinct origins".  The air has the same origin, but the OCS and O3 enrichment has different sources.**
*The air showing OCS concentrations over 550 ppt was transported from the North at high altitude (800-1000m) and was entrained downwards during the morning hours of June 2013. The air richer in O$_3$ (photosmog of the city of Marseille) which is advected in the afternoon above the OHP site is transported at low altitude from the S-SW by the sea breeze. The air masses have distinct origins.*

**Page 9, line 23: "Backward trajectories at 300 m above ground level ending at 12 UTC, when OCS levels at the O3HP in June 2013 were over 600 ppt, show that the circulation of air masses during both periods was at low altitude...". Define "both periods". Do the authors mean 2012 and 2013? The sentence before points to only 2013 data.**
*Yes, we meant 2012 and 2013.*
*Taken into account in the revised MS.*

**Page 9, line 24-30:  It is not clear here what message the authors try to convey.  The authors point out two different trajectories: one is from the Rhône Valley, where anthropogenic emissions could cause a rise in OCS. The other is the sea breeze, which (I presume) could transport the high O3 concentrations from the Marseille area, but this peak does not coincide with the OCS peak.**
*The air showing OCS concentrations over 550 ppt was transported from the North at high altitude (800-1000m) and was entrained downwards during the morning hours of June 2013. The air richer in*

*O$_3$ (photosmog of the city of Marseille) which is advected in the afternoon above the OHP site is transported at low altitude from the S-SW by the sea breeze. The air masses have distinct origins.*

**Page 9, line 28-29: The authors state that polluted air from the Marseille area is transported by a sea breeze, leading to an increase of ozone above the boundary layer. Why would a sea breeze cause transport above the boundary layer? I would say this transport happens within the boundary layer as a sea breeze causes horizontal movement from the sea towards land. Please also show why it is plausible that there is a sea breeze, did the wind direction change? Why would the Marseille area cause an ozone peak and not the Rhône Valley? And did ozone correlate with CO for the 2013 data?**
*This point is discussed above.*

**Page 10, line 10: the authors probably mean to refer to Fig. 5 instead of 3.**
*Correct. Sorry for that.*

**Page 10, line 14: ERU is defined as the ratio of the relative drawdown of OCS to CO2. Only when the plant uptake is the dominant flux, the ERU is proportional to the ratio of GPP/NEE with a proportionality constant that is the LRU (Campbell et al., 2008). Please make clear that the formulation that the authors use is only valid when the plant uptake is the dominant flux. After that the authors can explain that this is only the case at the OH3P site for a few hours in the afternoon (because at other moments the ecosystem is not the main driver but rather the boundary layer dynamics) and that ERU could only be calculated using the OCS and CO2 gradients for these few hours. Please give the numbers reported by Blonquist et al. (2011). I am also aware of ERU values presented by Maseyk et al. (2014). What do these ERU values tell us about the plant uptake? (Like the authors state in the third reason given in the beginning of section 4.3, see also my next comment).**

*This paragraph was rewritten as follows with your help:*
*"With these caveats in mind, the ratio of the mean relative vertical gradients of OCS and CO2 (calculated from linear OCS profiles) was equal to 4.7 and 4.3 for the afternoons of June 6 and 17 with, however, large relative error (≥ 50%), and was consistent with ERUs reported by Blonquist et al. (2011) at the Harvard Forest AmeriFlux site in summer-autumn 2006 (5.7 ± 1.2 (1 SD) for short-term ERU values calculated from linear OCS profiles as we did at the O3HP).*
*Only when the plant uptake is the dominant flux, the ERU is proportional to the ratio of GPP/NEE with a proportionality constant that is the LRU (Campbell et al., 2008). As discussed above, this is only the case at the O3HP site for a few hours in the afternoon (because at other moments the ecosystem is not the main driver but rather the boundary layer dynamics) and that ERU could only be calculated using the OCS and CO2 gradients for these few hours. When ERUs and the mean NEE/GPP ratio calculated for the period 12-17 UTC (0.78 ± 0.05, n=20) are used in Eq. 2, LRUs at the O3HP are equal to 3.7 and 3.4. These values fall in the upper range of LRUs obtained from leaf chamber studies over a large range of light conditions and tree species (1-4, Stimler et al., 2010; 1.3-2.3, Berkelhammer et al., 2014)."*

**Page 10, line 26-28: please clarify all three reasons to reach the conclusion that OCS uptake is the only relevant biospheric flux. This is not clear yet.**

*You are right. This paragraph was removed and replaced by the following one:*
*"Our OCS measurements were carried out during the period of maximum gross primary productivity of Mediterranean oak forests (Allard et al., 2008; Maselli et al., 2014). At the O3HP, the maximum of Q. pubescens net photosynthetic assimilation also occurs in June (Genard-Zielinski et al., in prep). The O3HP site appears to be ideal for the use of OCS uptake by plant as a tracer for GPP in a*

*Mediterranean oak forest because the soil is neither a source nor a sink of OCS when GPP fluxes culminate."*

**Page 11, line 31-33: Please rephrase, it reads as if the authors refer to the difference between the three open oak woodlands. But the authors probably mean the difference between these woodlands and the O3HP site. Also be more explicit how this conclusion is obtained: "The fact that no large nighttime drop of OCS is observed at O3HP suggests that the soil is not a net sink of OCS." The soil temperature and moisture have not changed from 2012 to 2013, and a early morning drawdown was indeed observed in 2012.**

*Indeed we meant "difference between these woodlands and the O3HP site". The sentence was corrected as suggested.*
*The process responsible for the early morning drawdown is photosynthesis not uptake by soil.*

**Page 12, line 10. Remove "If"**
*Done*

**Conclusions and perspectives: Page 13, line 15. Which requirements? Introduce them in the introduction and repeat here. Did the authors refer to the spring in 2012 only?**
**Page 13, line 15-17: The authors state that the soil uptake of OCS is negligible compared to the uptake of this gas through the stomata, however, I think this conclusion is made too easily. In fact no net exchange of OCS during the night is observed, which could either mean that there is no soil and leaf flux during the night, or that the sources and sinks (either from the soil or leaves) compensate each other. State clearly that this is just a speculation.**

*The conclusion has been largely rewritten as follows:*
*"Diel changes in OCS mixing ratio and in its vertical distribution show that net soil exchange of OCS is negligible compared to the uptake of this gas through the stomata, a feature which is not shared by other oak woodland ecosystems characterized by a Mediterranean climate. Hence, O3HP would be the adequate place to support the installation in the Mediterranean region of a monitoring station of OCS uptake by plants from eddy covariance measurements. However, the assessment of GPP from measured OCS fluxes at the ecosystem scale remains tributary of our poor knowledge of LRU diel variations at the O3HP which requires further examination using new experimental facilities (branch chambers or bags and/or coupled NEE/ERU measurements). In the framework of the European infrastructure Integrated Carbon Observation System (ICOS), an atmospheric measurement station (100 m high tower) has been set up at OHP in the year 2014 to determine multi-year records of greenhouse gases. Future research on the ERU is encouraged by the site being suitable to perform continuous and high precision vertical profiles of OCS using quantum cascade laser spectrometry. Unfortunately, our preliminary surveys suggest that the site is less adequate for estimating GPP from observations of vertical gradients of OCS relative to $CO_2$ during daytime than from eddy covariance measurements; the time window for calculation of the ecosystem relative uptake of OCS was found to be restricted at the O3HP to few hours after midday (1) because the vertical distribution of OCS is disrupted by entrainment in the morning of OCS rich tropospheric air sometimes contaminated by anthropogenic emissions, and (2) because the $CO_2$ vertical gradient reverses when it is still light."*

**Page 13, line 21: which "second method" do the authors mean? Which is the first?**
*See above.*

**Page 13, line 19-22: The authors discuss here that LRU is needed to derive GPP from OCS fluxes, and then continue saying that there were difficulties in determining ERU. To my knowledge LRU can only be derived from leaf-level measurements with branch chamber/bag measurements (e.g. Berkelhammer et al., 2014), how do your perspectives tackle the issue of getting LRU values?**

*See above. We believe that coupled ERU/NEE measurements could help assessing the diel variations in LRU at OHP.*

**Figures**

**Fig 2. 2012 data are shown, but the 2013 data are at least as important due to the high afternoon OCS peaks. I suggest the authors show both the 2012 and 2013 data. Also interesting to see would be the wind direction as an indication for a sea breeze and H2O as indication for entrainment.**

*This would indeed be possible for OCS because the 2012 and 2013 data measured by GC were calibrated but $CO_2$ and CO data measured in 2013 with the LGR instrument were not calibrated. We think that it is not right to compare calibrated data with raw data. New figures have been prepared to show the wind direction as an indication for the sea breeze and $H_2O$ as indication for entrainment. They are available in the supplementary information.*

**Fig 3a. This can already be seen from Fig 2c and 2d. I suggest the authors include meteo and concentration data of 2013 in Fig 2, then remove Fig 3, and include the average daily cycle of ozone in Fig 4 (to still be able to make the comparison between OCS and ozone).**

*Figure 3 is important because it shows the fine scale variations of OCS and $O_3$ and the periods during which these variations are in phase and out of phase. Data shown in Fig. 4 were not corrected for any "long-term" trend. To include an average daily cycle of ozone in Fig. 4, it would be necessary to correct data of 2012 for the clear increasing trend in $O_3$ which was less noticeable in the 2013 record. In fact, if revised as suggested by the reviewer, Fig. 4 would compare the mean diel patterns in ambient OCS mixing ratios to the mean diel patterns in $O_3$ anomalies. We have decided to keep the figures 3 and 4 roughly unchanged but added a night/day shading in Fig. 3 as suggested by reviewer 2.*

**Fig 5. Please show uncertainty bars for OCS as for CO2.**

*Done.*